# Influence of sensory modality and control dynamics on human path integration

Akis Stavropoulos[1]*[†], Kaushik J Lakshminarasimhan[2†], Jean Laurens[3], Xaq Pitkow[4,5,6]*, Dora E Angelaki[1,5,7]*

[1]Center for Neural Science, New York University, New York, United States; [2]Center for Theoretical Neuroscience, Columbia University, New York, United States; [3]Ernst Strüngmann Institute for Neuroscience, Frankfurt, Germany; [4]Department of Electrical and Computer Engineering, Rice University, Houston, United States; [5]Department of Neuroscience, Baylor College of Medicine, Houston, United States; [6]Center for Neuroscience and Artificial Intelligence, Baylor College of Medicine, Houston, United States; [7]Tandon School of Engineering, New York University, New York, United States

**Abstract** Path integration is a sensorimotor computation that can be used to infer latent dynamical states by integrating self-motion cues. We studied the influence of sensory observation (visual/vestibular) and latent control dynamics (velocity/acceleration) on human path integration using a novel motion-cueing algorithm. Sensory modality and control dynamics were both varied randomly across trials, as participants controlled a joystick to steer to a memorized target location in virtual reality. Visual and vestibular steering cues allowed comparable accuracies only when participants controlled their acceleration, suggesting that vestibular signals, on their own, fail to support accurate path integration in the absence of sustained acceleration. Nevertheless, performance in all conditions reflected a failure to fully adapt to changes in the underlying control dynamics, a result that was well explained by a bias in the dynamics estimation. This work demonstrates how an incorrect internal model of control dynamics affects navigation in volatile environments in spite of continuous sensory feedback.

## Editor's evaluation

This paper investigates the importance of visual and inertial sensory cues as well as the underlying motion dynamics to the accuracy of spatial navigation. When motion control was artificially manipulated in a virtual environment, subjects could navigate accurately using vision, but not inertial signals alone. Overall, these findings shed new light on how the brain combines sensory information and internal models of control dynamics for self-motion perception and navigation.

## Introduction

Imagine driving a car onto an icy road, where steering dynamics can change rapidly. To avoid crashing, one must rapidly infer the new dynamics and respond appropriately to keep the car on the desired path. Conversely, when you leave an ice patch, control dynamics change again, compelling you to re-adjust your steering. The quality of sensory cues may also vary depending on environmental factors (e.g. reduced visibility in fog or twilight, sub-threshold vestibular stimulation under near-constant travel velocity). Humans are adept at using time-varying sensory cues to adapt quickly to a wide range of latent control dynamics in volatile environments. However, the relative contributions of different

*For correspondence:
ges6@nyu.edu (AS);
xaq@rice.edu (XP);
da93@nyu.edu (DEA)

[†]These authors contributed equally to this work

**Competing interest:** The authors declare that no competing interests exist.

sensory modalities and the precise impact of latent control dynamics on goal-directed navigation remain poorly understood. Here, we study this in the context of path integration.

Path integration, a natural computation in which the brain uses dynamic sensory cues to infer the evolution of latent world states to continuously maintain a self-position estimate, has been studied in humans, but past experimental paradigms imposed several constraints. First, in many tasks, the motion was passive and/or restricted along predetermined, often one-dimensional (1D), trajectories (*Klatzky, 1998*; *Jürgens and Becker, 2006*; *Petzschner and Glasauer, 2011*; *Campos et al., 2012*; *Tramper and Medendorp, 2015*). Second, unlike time-varying actions that characterize navigation under natural conditions, participants' responses were often reduced to single, binary end-of-trial decisions (*ter Horst et al., 2015*; *Chrastil et al., 2016*; *Koppen et al., 2019*). Third, even studies that explored contributions of different sensory modalities in naturalistic settings failed to properly disentangle vestibular from motor cues generated during active locomotion (*Kearns et al., 2002*; *Campos et al., 2010*; *Bergmann et al., 2011*; *Chen et al., 2017*; *Schubert et al., 2012*; *Péruch et al., 1999*; *Péruch et al., 2005*). Furthermore, varying constraints have presumably resulted in inconsistent findings on the contribution of vestibular cues to path integration (*Jürgens and Becker, 2006*; *Campos et al., 2010*; *ter Horst et al., 2015*; *Tramper and Medendorp, 2015*; *Koppen et al., 2019*; *Chrastil et al., 2019*; *Glasauer et al., 1994*; *Seidman, 2008*).

There is a tight link between path integration and spatial navigation on the one hand, and internal models and control dynamics on the other. To accurately estimate self-motion, we rely not only on momentary sensory evidence but also on the knowledge of motion dynamics, that is, an internal model of the world. Knowledge of the dynamics makes the sensory consequences of actions predictable, allowing for more dexterous steering. However, although there is a large body of research focused on dynamics and adaptation for motor control (*Shadmehr and Mussa-Ivaldi, 1994*; *Lackner and Dizio, 1994*; *Krakauer et al., 1999*; *Takahashi et al., 2001*; *Burdet et al., 2001*; *Kording et al., 2007*; *Berniker et al., 2010*), studies of perceptual inference of latent dynamics during navigation have been limited. Some pioneering studies demonstrated participants' ability to reproduce arbitrary 1D velocity profiles (*Grasso et al., 1999*; *Israël et al., 1997*), while more recent efforts showed that the history of linear (*Petzschner and Glasauer, 2011*) and angular (*Prsa et al., 2015*) displacements affects how participants process sensory input in the current trial. We previously observed that false expectations about the magnitude of self-motion can have a drastic effect on path integration (*Lakshminarasimhan et al., 2018*). We wondered whether prior expectations about the *temporal dynamics* of self-motion, that is, how velocities are temporally correlated, can also propagate over time to influence navigation.

To explore how dynamics influence navigation across sensory modalities (visual, vestibular, or both), we have built upon a naturalistic paradigm of path integration in which participants navigate to a briefly cued target location using a joystick to control their velocity in a virtual visual environment (*Lakshminarasimhan et al., 2018*; *Alefantis et al., 2021*). Here, we generalize this framework by varying both the control dynamics (joystick control varied along a continuum from velocity to acceleration) and the available sensory cues (vestibular, visual, or both). To achieve this, we designed a motion-cueing (MC) algorithm to render self-motion stimuli according to a joystick control input of maintained accelerations while maintaining correspondence between visual (optic flow) and inertial cues. Using a motion platform with six degrees of freedom to approximate the accelerations that an observer would feel under the imposed control dynamics, we ensured that the MC algorithm would generate matching visual and vestibular cues to closely approximate the desired self-motion (see Materials and methods, *Figure 1—figure supplements 1 and 2*). The development of the MC algorithm represents a departure from classical paradigms of navigation research in humans (*Chrastil et al., 2019*; *Israël et al., 1996*; *Koppen et al., 2019*; *Seemungal et al., 2007*; *ter Horst et al., 2015*), as it helps eliminate artificial constraints while still allowing for the isolation of different sensory contributions, most notably vestibular/somatosensory cues, during active, volitional, steering.

We found that participants' steering responses were biased (undershooting), and the biases were more prominent in the vestibular condition. Furthermore, steering biases were strongly modulated by the underlying control dynamics. These findings suggest that inertial cues alone (as generated by motion cueing) lack the reliability to support accurate path integration in the absence of sustained acceleration, and that an accurate internal model of control dynamics is needed to make use of sensory observations when navigating in volatile environments.

# Results

## Task structure

Human participants steered toward a briefly cued target location on a virtual ground plane, with varying sensory conditions and control dynamics interleaved across trials. Participants sat on a motion platform in front of a screen displaying a virtual environment (*Figure 1A*). Stereoscopic depth cues were provided using polarizing goggles. On each trial, a circular target appeared briefly at a random location (drawn from a uniform distribution within the field of view; *Figure 1B and C*) and participants had to navigate to the remembered target location in the virtual world using a joystick to control linear and angular self-motion. The virtual ground plane was defined visually by a texture of many small triangles which independently appeared only transiently; they could therefore only provide optic-flow information and could not be used as landmarks. The self-motion process evolved according to Markov dynamics, such that the movement velocity at the next time step depended only on the current joystick input and the current velocity (Materials and methods – *Equation 1a*).

A time constant for the control filter (control timescale) governed the control dynamics: in trials with a *small* time constant and a fast filter, joystick position essentially controlled velocity, providing participants with responsive control over their self-motion, resembling regular road-driving dynamics. However, when the time constant was *large* and the control filter was slow, joystick position mainly controlled acceleration, mimicking high inertia under viscous damping, as one would experience on an icy road where steering is sluggish (*Figure 1D* right and 1E – top vs. bottom). For these experiments, as the control timescale changed, the maximum velocity was adjusted so that the participant could reach the typical target in about the same amount of time on average. This design ensured that the effect of changing control dynamics would not be confused with the effect of integrating sensory signals over a longer or shorter time.

Concurrently, we manipulated the modality of sensory observations to generate three conditions: (1) a *vestibular* condition in which participants navigated in darkness, and sensed only the platform's motion (note that this condition also engages somatosensory cues, see Materials and methods), (2) a *visual* condition in which the motion platform was stationary and velocity was signaled by optic flow, and (3) a *combined* condition in which both cues were available (*Figure 1E* – left to right). Across trials, sensory conditions were randomly interleaved while manipulation of the time constant followed a bounded random walk (Materials and methods – *Equation 2*). Participants did not receive any performance-related feedback.

## Effect of sensory modality on performance

We first compared the participants' stopping locations on each trial to the corresponding target locations, separately for each sensory condition. We calculated the radial distance $\tilde{r}$ and angular eccentricity $\tilde{\theta}$ of the participants' final position relative to the initial position (*Figure 2A*), and compared them to the initial target distance $r$ and angle $\theta$, as shown for all trials (all time constants together) of a typical participant in *Figure 2B and C*. This revealed biased performance with notable under-shooting (participants stopped short of the true target location), in both distance and angle, which was well described by a linear model without intercept (radial distance $R^2$ ± standard deviation – vestibular: 0.39 ± 0.06, visual: 0.67 ± 0.1, combined: 0.64 ± 0.11; angular eccentricity $R^2$ ± standard deviation – vestibular: 0.85 ± 0.06, visual: 0.95 ± 0.05, combined: 0.96 ± 0.04. Adding a non-zero intercept term offered negligible improvement; radial distance $\Delta R^2$ – vestibular: 0.02 ± 0.02, visual: 0.03 ± 0.03, combined: 0.03 ± 0.02; angular eccentricity $\Delta R^2$ – vestibular: 0.02 ± 0.03, visual: 0.01 ± 0.01, combined: 0.01 ± 0.01). We refer to the slope of the linear regression as 'response gain': a response gain of unity indicates no bias, while gains larger (smaller) than unity indicate overshooting (undershooting). As shown with the example participant in *Figure 2B and C*, there was substantial undershooting in the vestibular condition, whereas performance was relatively unbiased under the combined and visual conditions (see also *Figure 2—figure supplement 1A*). These results were consistent across participants (*Figure 2D*, mean radial gain± standard deviation – vestibular: 0.76 ± 0.25, visual: 0.88 ± 0.23, combined: 0.85 ± 0.22, mean angular gain± standard deviation – vestibular: 0.79 ± 0.22, visual: 0.98 ± 0.14, combined: 0.95 ± 0.12), and no significant sex differences were observed (see *Figure 2—figure supplement 1B*). The difference in response gain between modalities could be traced back to the control exerted by the subjects on the joystick. Both linear and angular components of control input had shorter duration in the vestibular condition (mean ± SEM of total

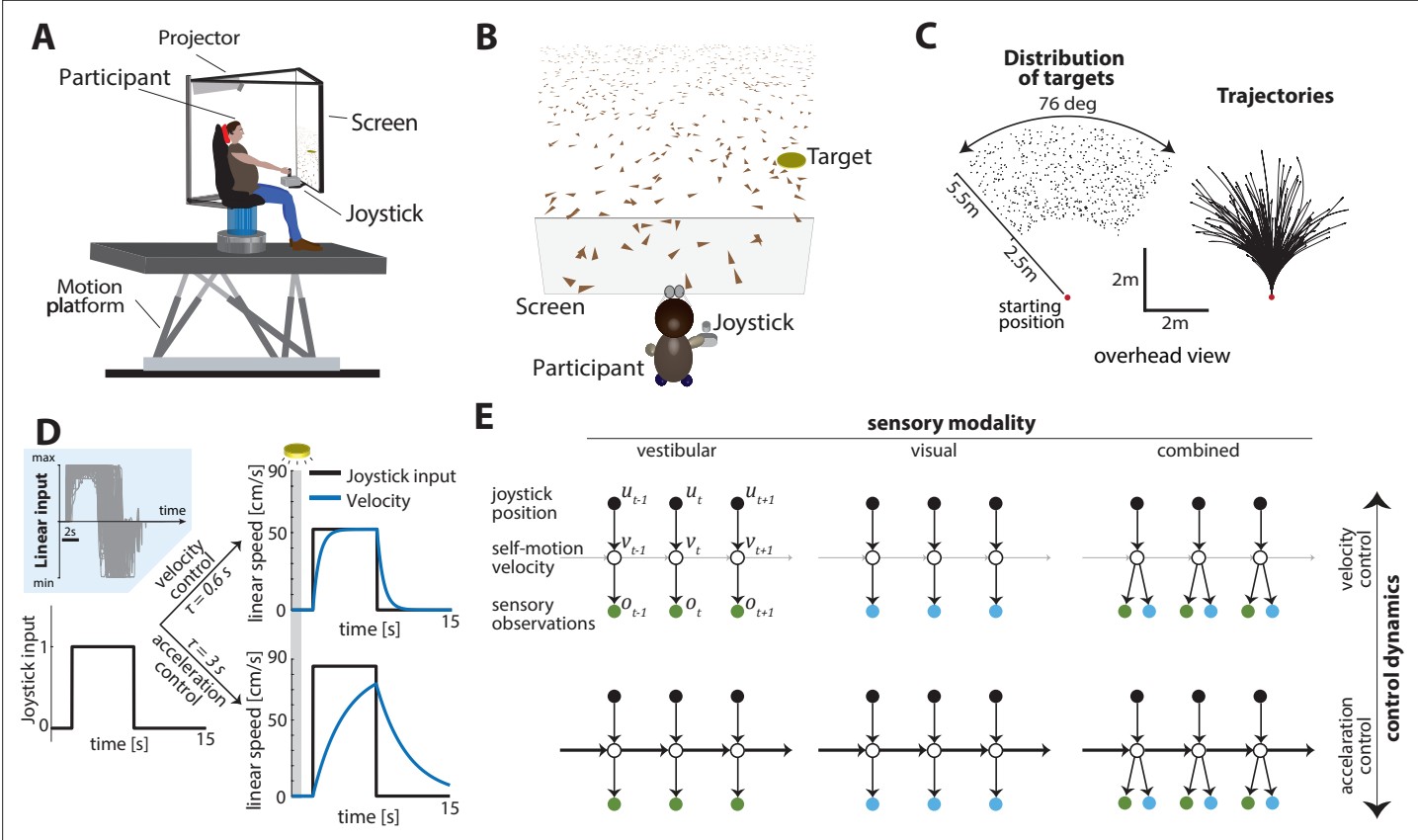

**Figure 1.** Experimental design. (**A**) Experimental setup. Participants sit on a six-degrees-of-freedom motion platform with a coupled rotator that allowed unlimited yaw displacements. Visual stimuli were back-projected on a screen (see Materials and methods). The joystick participants used to navigate in the virtual world is mounted in front of the participants' midline. (**B**) Schematic view of the experimental virtual environment. Participants use a joystick to navigate to a cued target (*yellow disc*) using optic-flow cues generated by ground plane elements (*brown triangles; visual and combined conditions only*). The ground plane elements appeared transiently at random orientations to ensure they cannot serve as spatial or angular landmarks. (**C**) Left: Overhead view of the spatial distribution of target positions across trials. *Red dot* shows the starting position of the participant. Positions were uniformly distributed within the participant's field of view. Right: Movement trajectories of one participant during a representative subset of trials. Starting location is denoted by the *red dot*. (**D**) Control dynamics. Inset: Linear joystick input from a subset of trials in the visual condition of an example participant. Left: Simulated maximum pulse joystick input (*max joystick input = 1*) (see also **Figure 1—figure supplement 3**). This input is lowpass filtered to mimic the existence of inertia. The time constant of the filter varies across trials (time constant *τ*). In our framework, maximum velocity also varies according to the time constant *τ* of each trial to ensure comparable travel times across trials (see Materials and methods – *Control Dynamics*). Right: The same joystick input (scaled by the corresponding maximum velocity for each *τ*) produces different velocity profiles for different time constants (*τ* = 0.6 s corresponds to *velocity control*; *τ* = 3 s corresponds to *acceleration control*; *τ* values varied randomly along a continuum across trials, see Materials and methods). Also depicted is the brief cueing period of the target at the beginning of the trial (gray zone, 1 s long). (**E**) Markov decision process governing self-motion sensation (Materials and methods – **Equation 1a**). *u*, *v*, and *o* denote joystick input, movement velocity, and sensory observations, respectively, and subscripts denote time indices. Note that due to the 2D nature of the task, these variables are all vector-valued, but we depict them as scalars for the purpose of illustration. By varying the time constant, we manipulated the control dynamics (i.e., the degree to which the current velocity carried over to the future, indicated by the thickness of the horizontal lines) along a continuum such that the joystick position primarily determined either the participant's velocity (top; *thin lines*) or acceleration (bottom; *thick lines*) (compare with (**D**) top and bottom, respectively). Sensory observations were available in the form of vestibular (left), optic flow (middle), or both (right).

The online version of this article includes the following figure supplement(s) for figure 1:

**Figure supplement 1.** Motion Cueing framework.

**Figure supplement 2.** Verification of Motion Cueing output.

**Figure supplement 3.** Control dynamics framework.

area of joystick input across participants (a.u.): radial – vestibular: 5.62 ± 0.27, visual: 7.31 ± 0.33, combined: 7.07 ± 0.34; angular – vestibular: 2.39 ± 0.30, visual: 3.29 ± 0.42, combined: 3.79 ± 0.46), and produced smaller displacements, as summarized by the response gains (**Figure 2**, **Figure 2— figure supplement 2**).

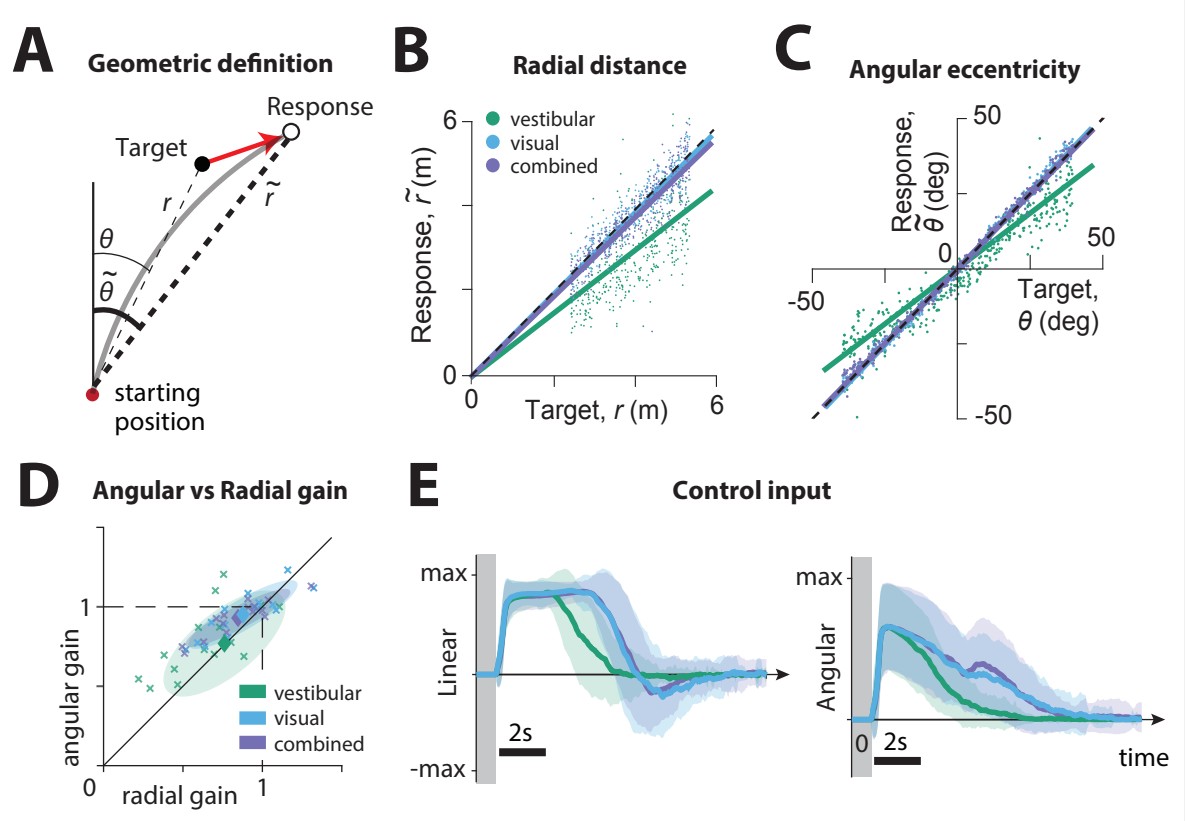

**Figure 2.** Effect of sensory modality on participants' responses. (**A**) Geometric definition of analysis variables. The *gray solid line* indicates an example trajectory. The target and response distance and angle relative to the starting position of the participant are given by $r, \theta$ (*thin lines*) and $\tilde{r}, \tilde{\theta}$ (*thick lines*), respectively. (**B, C**) Example participant: Comparison of the radial distance $\tilde{r}$ of an example participant's response (final position) against the radial distance $r$ of the target (**B**), as well as the angular eccentricity of the participant's response $\tilde{\theta}$ vs. target angle $\theta$ (**C**), across all trials for one participant, colored according to the sensory condition (*green: vestibular, cyan: visual, purple: combined visual and vestibular; Figure 2—source data 1*). Radial and angular response gains were defined as the slope of the corresponding regressions. Black dashed lines show unity slope, and the solid lines represent slopes of the regression fits (intercept set to 0). (**D**) All participants: Radial and angular gains in each sensory condition plotted for each individual participant (*Figure 2—source data 2*). *Ellipses* show 68% confidence intervals of the distribution of data points for the corresponding sensory condition. *Diamonds* (centers of the ellipses) represent the mean radial and angular response gains across participants. Dashed lines indicate unbiased radial or angular position responses. Solid diagonal line has unit slope. (**E**) Magnitudes of radial and angular components of control inputs across sensory conditions for an example participant. *Shaded regions* represent ±1 standard deviation across trials. The *gray zone* corresponds to the target presentation period.

The online version of this article includes the following source data and figure supplement(s) for figure 2:

**Source data 1.** Radial and angular responses.

**Source data 2.** Response gains.

**Figure supplement 1.** Trajectories in all conditions and sex differences in performance.

**Figure supplement 2.** Joystick inputs.

## Effect of control dynamics on performance

To examine whether control dynamics affected the response gain, we performed three complementary analyses. First, we recomputed response gains by stratifying the trials into three groups of equal size based on the time constants. We found that smaller time constants (velocity control) were associated with smaller response gains (*Figure 3A*; *Appendix 2—table 1*). This relationship was most pronounced in the vestibular condition, where larger time constants (acceleration control) resulted in better (closer to ideal) performance (*Figure 3*, green; see Discussion). Control dynamics had a smaller but considerable effect on steering responses in the visual and combined conditions, with participants exhibiting modest overshooting (undershooting) when the time constant was large (small) (*Figure 3A*, cyan/purple).

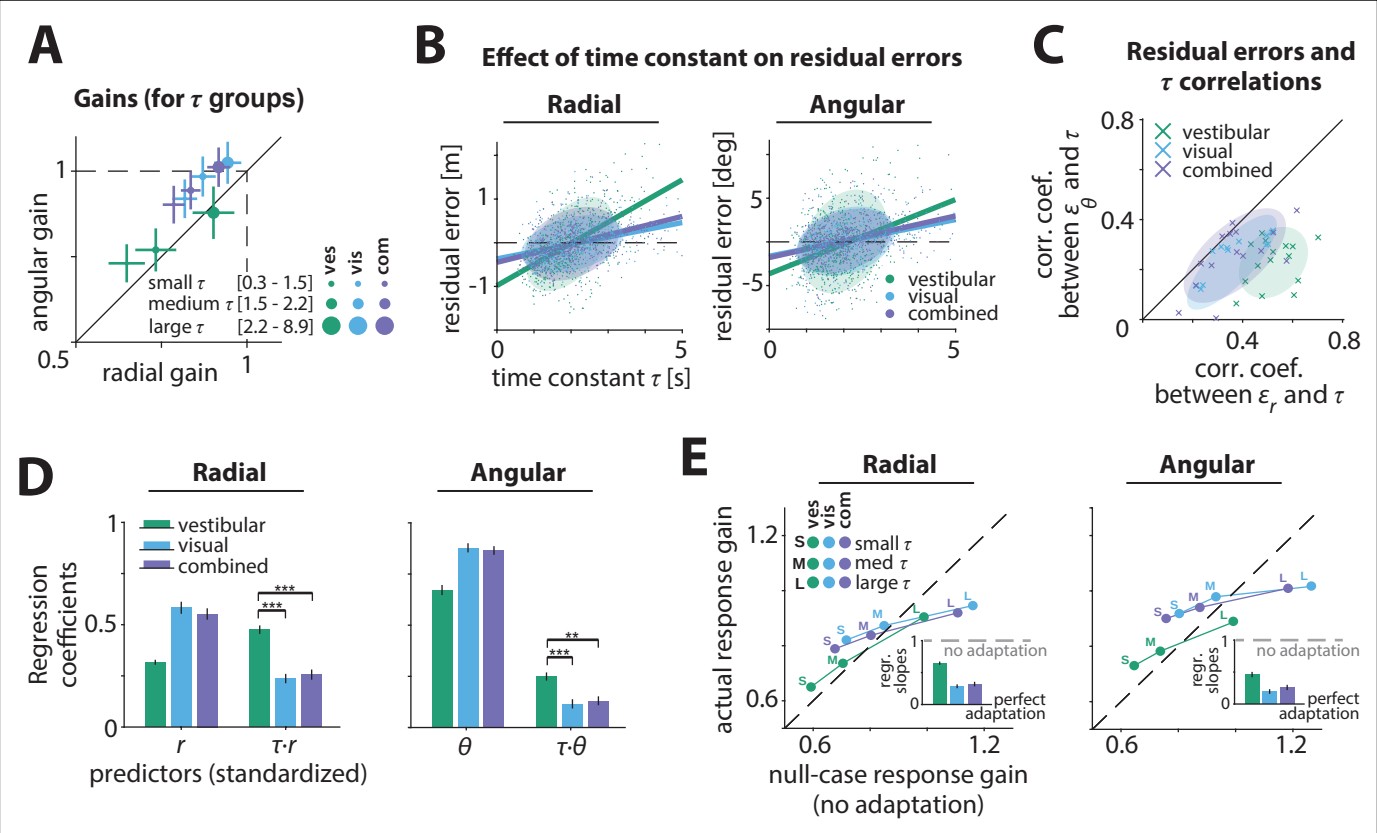

**Figure 3.** Effect of control dynamics on participants' responses. (**A**) Participant average of radial and angular response gains in each condition, with trials grouped into tertiles of increasing time constant $\tau$. Error bars denote ±1 SEM. (**B**) Effect of time constant $\tau$ on radial (left) and angular (right) residual error, for an example participant (*Figure 3—source data 1*). *Solid lines* represent linear regression fits and *ellipses* the 68% confidence interval of the distribution for each sensory condition. *Dashed lines* denote zero residual error (i.e. stopping location matches mean response). (**C**) Correlations of radial ($\varepsilon_r$) and angular ($\varepsilon_\theta$) residual errors with the time constant for all participants. *Ellipses* indicate the 68% confidence intervals of the distribution of data points for each sensory condition. Solid diagonal line has unit slope. Across participants, radial correlations, which were larger for the vestibular condition, were greater than angular correlations (see also *Appendix 2—table 2*). (**D**) Linear regression coefficients for the prediction of participants' response location (final position: $\tilde{r}$, $\tilde{\theta}$; left and right, respectively) from initial target location ($r, \theta$) and the interaction between initial target location and the time constant ($r \cdot \tau, \theta \cdot \tau$) (all variables were standardized before regressing, see Materials and methods; *Figure 3—source data 2*). *Asterisks* denote statistical significance of the difference in coefficient values of the interaction terms across sensory conditions (paired *t*-test; *: p<0.05, **: p<0.01, ***: p<0.001; *see* main text). *Error bars* denote ±1 SEM. Note a qualitative agreement between the terms that included target location only and the gains calculated with the simple linear regression model (*Figure 2B*). (**E**) Comparison of actual and null-case (no adaptation) response gains, for radial (*top*) and angular (*bottom*) components, respectively (average across participants). *Dashed lines* represent unity lines, that is, actual response gain corresponds to no adaptation. Inset: Regression slopes between actual and null-case response gains. A slope of 0 or 1 corresponds to perfect or no adaptation (*gray dashed lines*), respectively. *Error bars* denote ±1 SEM.

The online version of this article includes the following source data and figure supplement(s) for figure 3:

**Source data 1.** Residual errors.

**Source data 2.** Regression coefficients between time constant and residual errors.

**Figure supplement 1.** Effect of time constant on residual errors.

**Figure supplement 2.** Partial correlation analysis.

Second, we performed a fine-grained version of the above analysis by computing residual errors on each trial, that is, the deviation of the response from the mean response predicted from target location alone (Materials and methods – *Equation 3*). Since participants try to stop at their believed target location, ideally their mean responses should depend only on target location, and not on control dynamics. In other words, if participants adapted their control appropriately to the varying control dynamics, their responses should cluster around their mean response, and as a result, their residual errors should be centered around zero without any mean dependence on dynamics. However, we found a significant correlation between residual errors and the time constant across trials (*Figure 3B,*

*C*; *Figure 3—figure supplement 1*, *Appendix 2—table 2*, see Materials and methods; no significant sex differences were observed, and therefore are not investigated in subsequent analyses, see also *Figure 2—figure supplement 1C*). This correlation and the corresponding regression slopes were substantially higher in the vestibular condition (mean Pearson's *r* ± SEM: radial component – vestibular: 0.52 ± 0.02, visual: 0.36 ± 0.03, combined: 0.37 ± 0.03; angular component – vestibular: 0.23 ± 0.02, visual: 0.23 ± 0.03, combined: 0.26 ± 0.03; see also *Appendix 2—Tables 2 and 3*). Thus, for a given target distance, participants tended to travel further when the time constant was larger (acceleration control), indicating they did not fully adapt their steering control to the underlying dynamics.

Third, to quantify the contribution of the time constant in the participants' responses, we expanded the linear model to accommodate a dependence of response (final stopping position) on target location, time constant, and their interaction. A partial correlation analyses revealed that the time constant contributed substantially to participants' response gain, albeit only by modulating the radial and angular distance dependence (*Appendix 2—table 4*; *Figure 3—figure supplement 2*; see Materials and methods – *Equation 4*). Again, the contribution of the time constant-dependent term was much greater for the vestibular condition (*Figure 3D*), especially for the radial distance (p-values of difference in coefficient values across modalities obtained by a paired *t*-test – radial: vestibular vs. visual: p < 10⁻⁴ , vestibular vs. combined: p < 10⁻⁴; angular: vestibular vs. visual: p = 0.016, vestibular vs. combined: p = 0.013). While perfect adaptation should lead to response gain that is independent of control dynamics, all three independent analyses revealed that control dynamics did substantially influence the steering response gain, exposing participants' failure to adapt their steering to the underlying dynamics. Adaptation was lowest for the vestibular condition; in contrast, for the visual and combined conditions, the response gain was less affected indicating greater compensation when visual information was available.

We quantified the extent to which participants failed to adapt to the control dynamics, by simulating a null case for no adaptation. Specifically, we generated null-case trajectories by using the steering input from actual trials and re-integrating it with time constants from other trials. In this set of null-case trajectories, the steering control corresponds to different time constants; in other words, steering is not adapted to the underlying dynamics (see Materials and methods). We then grouped these trajectories based on the simulation time constant (as in *Figure 3A*) and computed the corresponding response gains. We found that the true response gains in the vestibular condition were much closer to the no-adaptation null case, compared to visual/combined conditions (*Figure 3E*). Interestingly, this finding was more prominent in the radial component of the response gain (*Figure 3E*

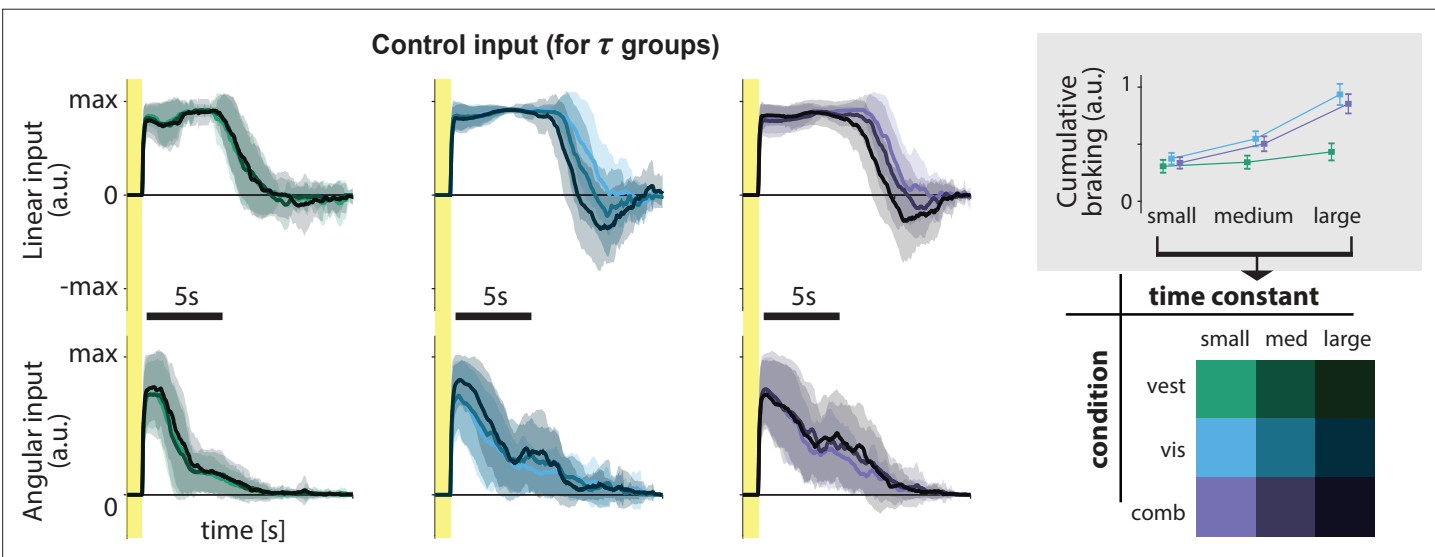

**Figure 4.** Linear and angular control inputs for each condition grouped based on the time constant (see *legend*; bottom right), for an example participant. *Shaded regions* represent ±1 standard deviation across trials. *Yellow zones* denote target presentation period. Inset: Cumulative braking (i.e. absolute sum of negative linear input) for each condition across time constant groups. Braking was averaged across trials. *Error bars* denote ±1 SEM across participants.

insets), consistent with our earlier observations of a stronger influence of the dynamics on the radial component of the responses.

We have shown how various measures of the participants' final responses (stopping positions, response gain, residual errors) are influenced by the time constant of the dynamics. This large dependence of the final responses on the time constant exposes participants' failure to fully adapt their steering to the underlying dynamics. In other words, the influence of the dynamics on steering control was relatively weak, especially in the vestibular condition.

For best performance, however, control dynamics *should* influence the time course of steering behavior. We directly quantified the influence of the control dynamics on steering by comparing participants' braking (negative control input) across time constants: when the time constant is large, we ideally expect to see more braking as a countermeasure for the sluggish control (*Figure 1D*) to minimize travel duration (see Materials and methods). Indeed, participants do tend to brake more for higher time constants, but this effect is weaker in the vestibular condition (*Figure 4 and* inset). Nevertheless, correlations between the time constant and cumulative braking (total area below zero linear control input) were significant in all sensory conditions (mean Pearson's *r* ± SEM – vestibular: 0.20 ± 0.03, visual: 0.62 ± 0.04, combined: 0.57 ± 0.04; p-values of Pearson's *r* difference from zero – vestibular: p = $10^{-5}$, visual: p < $10^{-7}$, combined: p < $10^{-7}$). Overall, it appears that behavior in the vestibular condition is minimally influenced by the dynamics (i.e. smaller modulation of control input by the time constant, as shown by the cumulative braking). When optic flow is available, however, participants are more flexible in adjusting their control.

We have shown previously that accumulating sensory noise over an extended time (~10 s) would lead to a large uncertainty in the participant's beliefs about their position, causing them to undershoot (*Lakshminarasimhan et al., 2018*). The exact amount of undershooting depends both on the reliability of self-motion cues, which determines the *instantaneous* uncertainty in the self-motion estimate, and on travel duration, which governs how much uncertainty is *accumulated* while navigating to the target. With recent findings ascribing uncertainty accumulation to noise in the velocity input (*Stangl et al., 2020*), the observed differences in navigation performance across sensory modalities can be readily attributed to greater measurement noise (lower reliability) in vestibular signals. On the other hand, we observed performance differences across control dynamics within each sensory modality, so those differences cannot be attributed to differences in the reliability of self-motion cues (*instantaneous uncertainty*). However, it might seem that this effect of control dynamics must be due to either differences in travel duration or velocity profiles, which would both affect the *accumulated uncertainty*. We adjusted stimulus parameters to ensure that the average travel time and average velocity were similar across different control dynamics (Materials and methods – *Equation 1.2-1.10*), however, we found that travel duration and average velocity depend weakly on the time constant in some participants. Simulations suggest that both dependencies are a consequence of maladaptation to the dynamics rather than a cause of the observed effect of the dynamics on the responses. Interestingly, the dependence is stronger in the vestibular condition where there is less adaptation to the dynamics, agreeing with our simulations (*Figure 5—figure supplement 1A, B*). Difference in velocity profiles is also an unlikely explanation since their expected effect on the participants' responses (undershoot) is the opposite of the observed effect of the control dynamics (overshooting tendency; *Figure 5—figure supplement 1C*). Consequently, unlike the effect of sensory modality on response gain, neither instantaneous nor accumulated differences in the uncertainty can fully account for the influence of control dynamics, that is, the time constant. Instead, we will now show that the data are well explained by strong prior expectations about motion dynamics that cause a bias in estimating the time constant.

## Modeling the effect of control dynamics across sensory modalities

From a normative standpoint, to optimally infer movement velocity, one must combine sensory observations with the knowledge of the time constant. Misestimating the time constant would produce errors in velocity estimates, which would then propagate to position estimates, leading control dynamics to influence response gain (*Figure 5A*, middle-right). This is akin to misestimating the slipperiness of an ice patch on the road causing an inappropriate steering response, which would culminate in a displacement that differs from the intended one (*Figure 5—figure supplement 2*). However, in the absence of performance-related feedback at the end of the trial, participants would be unaware of this discrepancy, wrongly believing that the actual trajectory was indeed the intended one. In other

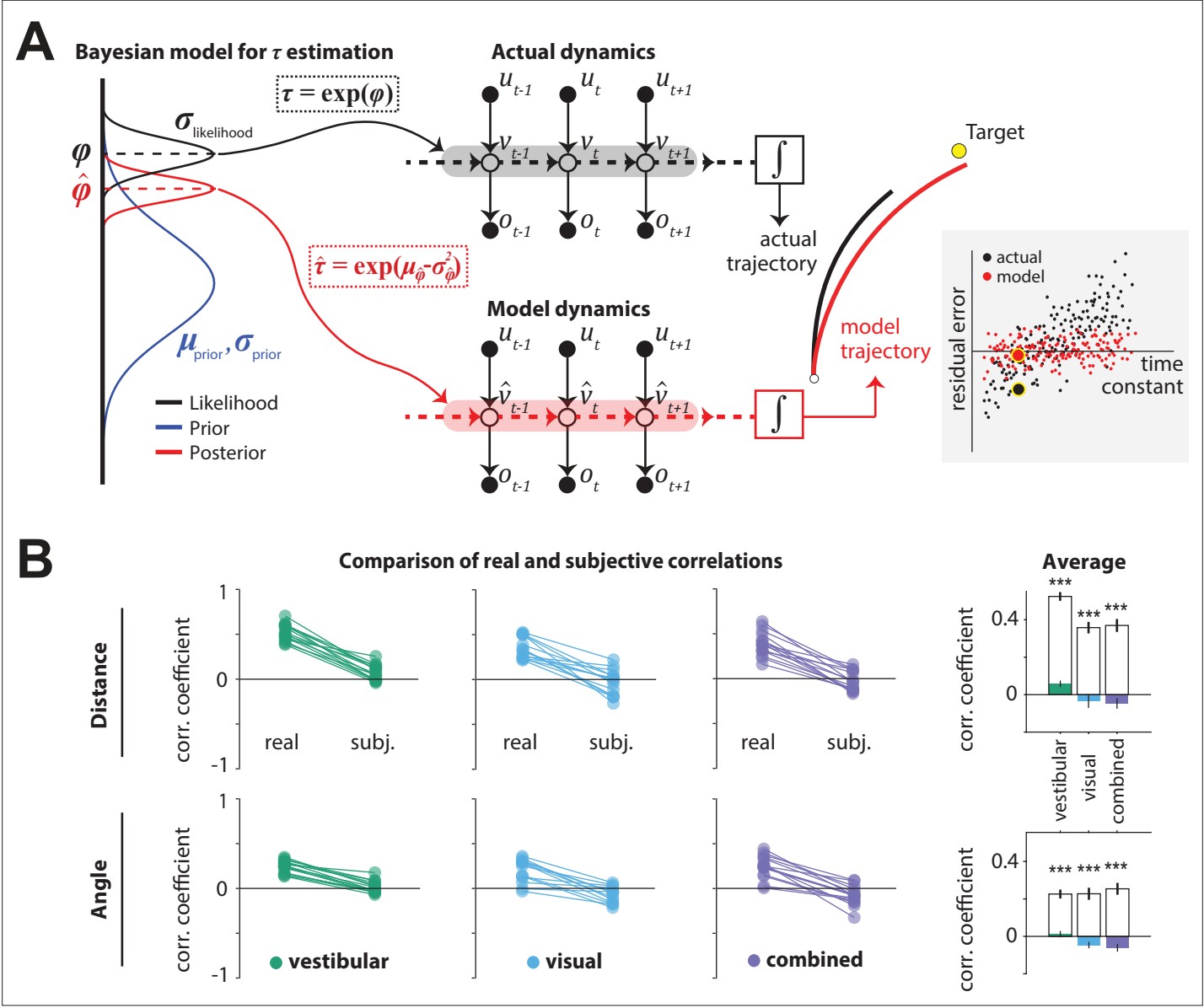

**Figure 5.** Bayesian model framework and correlations between the time constant and model-implied residual errors. (**A**)Left: Illustration of the Bayesian estimator model. We fit two parameters: the ratio $\lambda$ of standard deviations of prior and likelihood ($\lambda = \sigma_{\text{prior}}/\sigma_{\text{likelihood}}$) and the mean of the prior ($\mu_{\text{prior}}$) of the normally distributed variable $\varphi = \log \tau$ (*black dotted box*). Likelihood function is centered on the log-transformation of the actual $\tau$, $\varphi^* = \log \tau^*$ (*black dashed line*). The time constant estimate $\hat{\tau}$ corresponded to the median of the posterior distribution over $\tau$, which corresponds to the median $\hat{\varphi}$ over $\varphi$, $\hat{\tau} = \exp(\hat{\varphi})$ (*red dotted box;red dashed line*; see Materials and methods). Middle: Control dynamics implied by the actual time constant $\tau$ (top; gray shade) and the estimated time constant $\tau$ (bottom; red shade). $u$, $v$, and $o$ denote joystick input, movement velocity, and sensory observations, respectively, and subscripts denote time indices. $v$ denotes the inferred velocity implied by the model. Misestimation of the time constant leads to erroneous velocity estimates about self-motion $v$ which result in biased position beliefs. Right: Illustration of the actual (black) and believed (red) trajectories produced by integrating (box) the actual velocity $v$ and the estimated velocity $v$, respectively. *White and yellow dots* denote the starting and target position, respectively. Inset: Illustration of correlated (*black* dots) and uncorrelated (*red* dots) residual errors with the time constant for actual and model-implied responses (simulated data). For simplicity, we depict residual errors as one-dimensional and assume unbiased responses (response gain of 1). Blown-up dots with yellow halo correspond to the actual and model-implied trajectories of the right panel. *Solid black horizontal line* corresponds to zero residual error (i.e. stop on target location). (**B**) Comparison of correlations between real and subjective residual errors with $\tau$ (*Figure 5—source data 1*). On the right, participant averages of these correlations are shown. Colored bars: 'Subjective' correlations, open bars: Actual correlations. Error bars denote ±1 SEM across participants. *Asterisks* denote the level of statistical significance of differences between real and subjective correlations (*: $p<0.05$, **: $p<0.01$, ***: $p<0.001$).

The online version of this article includes the following source data and figure supplement(s) for figure 5:

*Figure 5 continued on next page*

*Figure 5 continued*

**Source data 1.** correlations between time constant and model-implied residual errors.

**Figure supplement 1.** Testing model assumptions.

**Figure supplement 2.** Changes in travel distance for a given control input under different control dynamics.

words, participants' imperfect adaptation to changes in control dynamics could be a consequence of control dynamics misestimation.

We tested the hypothesis that participants misestimated the time constant using a two-step model that reconstructs the participants' believed trajectories according to their *point estimate* of the time constant $\tau$, as follows. First, a Bayesian observer model infers the participant's belief about $\tau$ on individual trials, that is, the subjective posterior distribution over the time constant ($\tau$ inference step; *Figure 5A*, left). Second, we used the median of that belief to reconstruct the believed trajectory by integrating the actual joystick input according to the *estimated* time constant on that trial (integration step), resulting in a believed stopping location (*Figure 5A*, middle-right). In the absence of bias (response gain of one), the believed stopping locations should land on or near the target. However, various unmeasurable fluctuations in that belief across trials should lead to variability clustered around the target location. When the behavior is biased (response gain different from one, as was the case here – *Figure 2D*), this cluster should instead be centered around the participants' *mean* belief for that target location (determined from their biased responses and henceforth referred to as *mean stopping location*). Since the participants' goal is to stop as close to their perceived target location as possible, the deviation of believed stopping locations from the mean stopping location for a given target should be small. We call this deviation the *subjective* residual error. Therefore, we inferred the parameters of the Bayesian model separately for each participant by minimizing the *subjective* residual errors induced by the control dynamics using the principle of least squares (see Materials and methods for further details). We next describe the parameters of the Bayesian model and then describe the results of fitting the model to our data.

Because the time constant $\tau$ is always positive, we model both the prior distribution and the likelihood function over the variable $\varphi = \log \tau$ as Gaussians in log-space. We parameterized both the prior and the likelihood with a mean ($\mu$) and standard deviation ($\sigma$). The mean of the prior ($\mu$) was allowed to freely vary across sensory conditions but assumed to remain fixed across trials. On each trial, the likelihood was assumed to be centered on the actual value of the log time constant $\tau^*$ on that trial according to $\mu = \varphi^* = \log \tau^*$ and was therefore not a free parameter. Finally, we set the ratio $\lambda$ of prior over likelihood $\sigma$, to freely vary across sensory conditions. Thus, for each sensory condition, we fit two parameters: the $\mu$ of the prior, and the ratio ($\lambda$) of prior $\sigma$ to likelihood $\sigma$. As mentioned above, we fit the model to minimize the difference between their believed stopping locations and their experimentally measured mean stopping location (*subjective* residual errors), using a least-squares approach (Materials and methods) and obtained one set of parameters for each condition. Finally, the participant's estimated time constant $\tau$ on each trial was taken to be the median of the best-fit model, which equals the median of the distribution over $\varphi$ (*Figure 5A*, left). By integrating the subject's joystick inputs on each trial using $\tau$ rather than the actual time constant $\tau$, we computed the believed stopping location and the subjective residual errors implied by the best-fit model.

We then compared the correlations between the time constant and the residual errors for *real responses* (from data in *Figure 3B and C*) or *subjective* responses (from model), separately for radial and angular components. Because participants try to stop at their believed target location, the believed stopping position should depend only on target location and not on the control dynamics. Any departure would suggest that participants *knowingly* failed to account for the effect of the control dynamics, which would manifest as a dependence of the *subjective* residual errors on the time constant $\tau$. In other words, a good model of the participants' beliefs would predict that the *subjective* residual errors should be uncorrelated with the time constant $\tau$ (*Figure 5A* inset – red) even if the *real* residual errors are correlated with the time constant (*Figure 5A* inset – black). In all cases, we observed that the correlation between residual error and time constant was indeed significantly smaller when these errors were computed using the *subjective* (believed) rather than *real* stopping location (*Figure 5B*). In fact, subjective residual errors were completely uncorrelated with the time constant suggesting that the Bayesian model is a good model of participants' beliefs, and that the apparent influence of

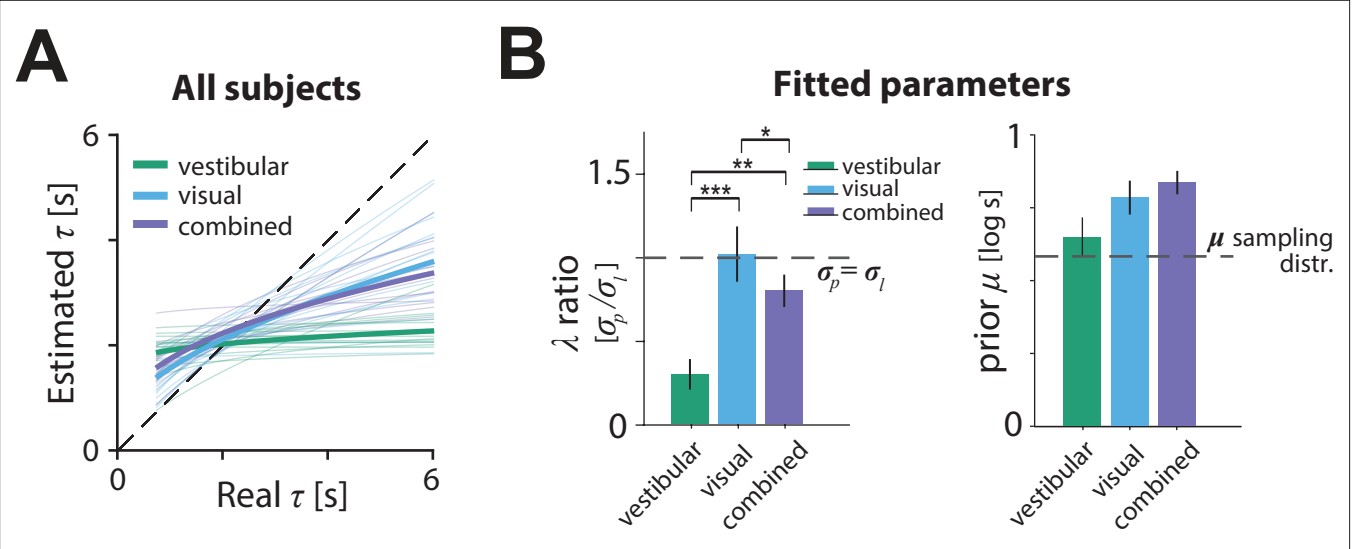

**Figure 6.** Model parameters. (**A**) Relationship between the model-estimated and actual time constant across all participants in vestibular (green), visual (cyan), and combined (purple) conditions. Participant averages are superimposed (*thick lines*). *Dashed line*: unbiased estimation (***Figure 6—source data 1***). (**B**) Fitted model parameters: ratio $\lambda$ of prior ($\sigma_p$) over likelihood ($\sigma_l$) standard deviation and mean ($\mu$) of prior. Error bars denote ±1 SEM. *Dashed lines* represent the corresponding values of the sampling distribution of $\varphi = \log \tau$, which is normal (see Materials and methods; ***Figure 6—source data 2***). The prior distribution's $\mu$ was comparable in the vestibular condition to the $\mu$ of the actual sampling distribution (sampling distribution $\mu$: 0.58 log *s* – p-value of prior $\mu$ difference obtained by bootstrapping – vestibular: p = 0.014, visual: p =< $10^{-7}$; combined: p < $10^{-7}$). *Asterisks* denote the level of statistical significance of differences in the fitted parameters across conditions (*: p<0.05, **: p<0.01, ***: p<0.001).

The online version of this article includes the following source data for figure 6:

**Source data 1.** Real versus estimated time constants.

**Source data 2.** Model parameters.

control dynamics on behavioral performance was entirely because participants misestimated the time constant of the underlying dynamics.

We next examined the model posterior estimates to assess how subjects' internal estimate of the control dynamics departed from the true dynamics. The relationship between real and model-estimated time constants for all participants can be seen in ***Figure 6A***. In the vestibular condition, all participants consistently misestimated $\tau$, exhibiting a substantial regression toward the mean (***Figure 6A***, green). This effect was much weaker in the visual condition. Only a few participants showed relatively flat estimates, with the majority showing smaller departures from ideal estimates (*dashed line*). The data for the combined condition followed a similar trend, with properties between those in the visual and vestibular conditions (***Figure 6A***, purple). These results suggest that the better control adaptation in the visual and combined conditions shown in ***Figure 3*** is due to participants' improved estimates of the time constant when optic flow was available.

The source of inaccuracies in the estimated time constant can be understood by examining the model parameters (***Figure 6B***).The ratio $\lambda$ of prior over likelihood standard deviations was significantly lower in the vestibular condition than other conditions, suggesting stronger relative weighting of the prior over the likelihood (***Figure 6B*** left, green bar; mean ratio $\lambda$± standard SEM – vestibular: 0.30 ± 0.09, visual: 1.02 ± 0.17, combined: 0.80 ± 0.10; p-value of ratio $\lambda$ paired differences obtained by boot-strapping – vestibular vs. visual: p = 0.0007, vestibular vs. combined: p = 0.0087; visual vs. combined: p = 0.016). Notably, the ratio was close to one for the visual and combined conditions, suggesting equal weighting of prior and likelihood. Thus, participants' estimate of the control dynamics in the vestibular condition was plagued by a combination of strong prior and weak likelihood, which explains the much stronger regression toward the mean in ***Figure 6A***.

## Alternative models

To test whether our assumption of a static prior distribution over time constants was reasonable, we fit an alternative Bayesian model in which the prior distribution was updated iteratively on every trial,

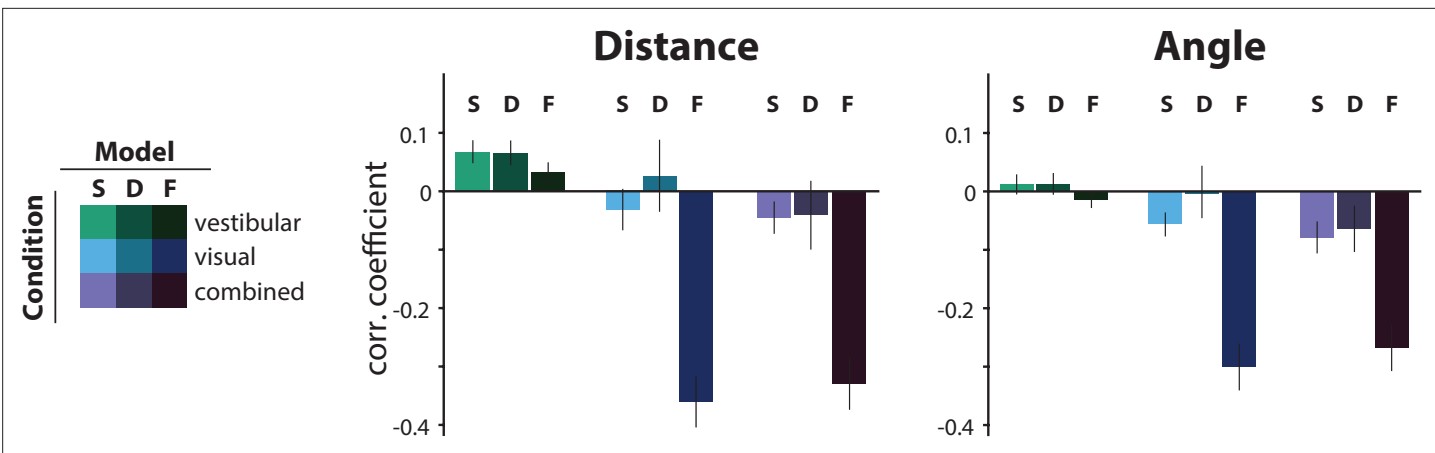

**Figure 7.** Comparison of the correlations between the actual τ and the subjective residual errors implied by three different τ-estimation models (Bayesian estimation with a static prior ([S], Bayesian estimation with a dynamic prior [D], fixed estimate [F]). We tested the hypotheses that either the prior distribution should not be static or that the participants ignored changes in the control dynamics and navigated according to a fixed time constant across all trials (fixed τ estimate model; see Materials and methods). For this, we compared the correlations between the subjective residual error and the actual trial τ that each model produces. The dynamic prior model performs similarly to the static prior model in all conditions, indicating that a static prior is adequate in explaining our data (p-values of paired t-test between correlation coefficients of the two models: distance – vestibular: p = 0.96, visual: p = 0.19, combined: p = 0.91; angle – vestibular: p = 0.87, visual: p = 0.09, combined: p = 0.59). For visual and combined conditions, the fixed τ model not only fails to minimize the correlations but, in fact, strongly reverses it, for both distance (*left*) and angle (*right*). Since these correlations arise from the believed trajectories that the fixed τ model produces, this suggests that participants knowingly stop before their believed target location for higher time constants. Model performance was only comparable in the vestibular condition, where the average correlation of the fixed τ model (F) was contained within the 95% confidence intervals (CI) of the static prior Bayesian model (*S*), for both distance and angle (distance – F: mean Pearson's correlation coefficient $\rho$ = 0.03, S: 95% CI of Pearson's correlation coefficient $\rho$ = [–0.10 0.25]; angle – F: mean Pearson's correlation coefficient $\rho$ = –0.01, S: 95% CI of Pearson's correlation coefficient $\rho$ = [–0.12 0.15]). Error bars denote ±1 SEM.

The online version of this article includes the following figure supplement(s) for figure 7:

**Figure supplement 1.** Correlation coefficients in the vestibular condition between the actual time constant and the subjective radial (left) and angular (right) residual errors, if participants carried over their τ estimate from the previous trial.

**Figure supplement 2.** Comparing bayesian and feedback control models.

as a weighted average of the prior on the previous trial and the current likelihood over $\varphi$ (Dynamic prior model; see Materials and methods). For this version, the initial prior $\mu$ was taken to be the time constant on the first trial, and we once again modeled the likelihood and prior as normal distributions over the log-transformed variable, $\varphi$, where the likelihood was centered on the actual $\varphi$ and was therefore not a free parameter. Thus, we fit one parameter: the ratio $\lambda$ of prior $\sigma$ over likelihood $\sigma$. On each trial, the relative weighting of prior and likelihood responsible for the update of the prior depended solely on $\lambda$; that is, the relationship between their corresponding $\sigma$ (i.e. relative widths). The performance of the static and dynamic prior models was comparable in all conditions, for both distance and angle, suggesting that a static prior is adequate in explaining the participants' behavior on this task (*Figure 7*; *light* vs. *dark* bars). In line with our expectations, when updating the prior in the dynamic model, the weighting of the previous-trial prior received significantly more weight in the vestibular condition (in the range of [0,1]; mean prior weights ± SEM– vestibular: 0.93 ± 0.03, visual: 0.48 ± 0.10, combined: 0.61 ± 0.09; p-value of paired weight differences obtained by boot-strapping – vestibular vs. visual: p = $10^{-5}$, vestibular vs. combined: p = $4 \cdot 10^{-4}$; visual vs. combined: p = 0.08). The comparable goodness of models with static and dynamic priors suggest that sensory observations were not too reliable to cause rapid changes in prior expectations during the course of the experiment.

At the other extreme, to test whether participants used sensory observations at all to estimate control dynamics, we compared the static prior Bayesian model to a parsimonious model that assumed a fixed time constant across all trials (i.e. completely ignoring changes in control dynamics). This latter model can be understood as a Bayesian model instantiated with a very strong static prior. In line with our expectations (see *Figure 6A*), this latter model performed comparably in the vestibular condition, but substantially worse in the visual and combined conditions (*Figure 7*).

Due to the correlated nature of the random walk process dictating the evolution of time constants, an alternative by which participants could get away without estimating the time constant in the vestibular condition would be to carry over their estimate from the previous combined/visual trial to the current vestibular trial. To test this, we considered two models: the time constant estimate in the current vestibular trial was taken to be either the real time constant or the posterior estimate of the time constant from the previous visual/combined trial. Neither model, however, could account for the participants' behavior, as they could not fully explain away the correlation between the residual errors and the time constant (*Figure 7—figure supplement 1*). Intuitively, although choosing actions with respect to the previous trial's time constant should result in estimates that regress toward the mean, the predicted effect is weaker than that observed in the data.

Finally, we tested a variation of previously suggested sensory feedback control models (*Glasauer et al., 2007*; *Grasso et al., 1999*) where a controller relies solely on sensory inputs to adjust their control without explicitly estimating the latent variables governing the control dynamics. Specifically, the model assumes that participants implement a type of bang-bang control that switches at a certain distance from the target (or more accurately, the mean response). However, this model predicts a much stronger dependence of the responses on the dynamics compared to our data, and characteristics of the predicted control input differ significantly from the actual control (*Figure 7—figure supplement 2*). Overall, our results suggest that optic flow, but not vestibular signals, primarily contributes to inferring the latent velocity dynamics.

## Discussion

We showed that human participants can navigate using different sensory cues and that changes in the control dynamics affect their performance. Specifically, we showed that participants can path integrate to steer toward a remembered target location quite accurately in the presence of optic flow. In contrast, inertial (vestibular/somatosensory) cues generated by motion cueing alone lacked the reliability to support accurate path integration, leading to substantially biased responses under velocity control. Performance was also influenced by the changing control dynamics in all sensory conditions. Because control dynamics were varied on a trial-by-trial basis, sensory cues were crucial for inferring those dynamics. We used probabilistic inference models to show that the observed responses are consistent with estimates of the control dynamics that were biased toward the center of the experimental distribution. This was particularly strong under the vestibular condition such that the response gain substantially increased as the motion dynamics tended toward acceleration control. Although control dynamics were correlated across trials, our models showed that participants did not take advantage of those correlations to improve their estimates.

### Relation to past work

In the paradigm used here, participants actively controlled linear and angular motion, allowing us to study multisensory path integration in two dimensions with few constraints. This paradigm was made possible by the development of an MC algorithm to render visual and vestibular cues either synchronously or separately. In contrast, previous studies on human path integration used restricted paradigms in which motion was either 1D or passively rendered, and participants' decisions were typically reduced to end-of-trial binary evaluations of relative displacement (*Campos et al., 2012*; *Chrastil et al., 2016*; *Chrastil et al., 2019*; *Jürgens and Becker, 2006*; *Koppen et al., 2019*; *ter Horst et al., 2015*; *Tramper and Medendorp, 2015*). As a result, findings from past studies that evaluate the contributions of different sensory modalities to self-motion perception (*Chrastil et al., 2019*; *Israël et al., 1996*; *Koppen et al., 2019*; *Seemungal et al., 2007*; *ter Horst et al., 2015*) may be more limited in generalizing to real-world navigation.

Our results show that, at least in humans, navigation is driven primarily by visual cues under conditions of near-constant travel velocity (velocity control). This dominance of vision suggests that the reliability of the visual cues is much higher than vestibular cues (as generated by our platform), as corroborated by the data from the combined condition in which performance resembles the visual condition. This makes sense because the vestibular system is mainly sensitive to acceleration, exhibiting higher sensitivity to higher-frequency motion compared to the visual system (*Karmali et al., 2014*). Consequently, it may only be reliable when motion is dominated by acceleration. This

interpretation is further supported by the observation that participants' vestibular performance was a lot less biased in the regime of acceleration joystick control, where accelerations are prolonged during navigation.

Experimental constraints in past navigation studies have also precluded examining the influence of control dynamics. In fact, the importance of accurately inferring control dynamics, which are critical for predicting the sensory consequences of actions, has largely been studied in the context of limb control and motor adaptation (*Burdet et al., 2001*; *Kording et al., 2007*; *Krakauer et al., 1999*; *Lackner and Dizio, 1994*; *Shadmehr and Mussa-Ivaldi, 1994*; *Takahashi et al., 2001*). Here, we provide evidence for the importance of accurately inferring control dynamics in the context of path integration and spatial navigation. Although participants were not instructed to expect changes in the latent dynamics and received no feedback, we showed that they nevertheless partly adapted to those dynamics while exhibiting a bias toward prior expectations about these dynamics. This biased estimation of control dynamics led to biased path integration performance. This result is analogous to findings about the effect of changing control dynamics in motor control: first, adaptation to the dynamics happens even in the absence of performance-related feedback (*Batcho et al., 2016*; *Lackner and Dizio, 1994*) and, second, this adaptation relies on prior experience (*Arce et al., 2009*) and leads to systematic errors when knowledge of the dynamics is inaccurate (*Körding et al., 2004*). Thus, participants try to exploit the additional information that the dynamics contain about their self-motion in order to achieve the desired displacement.

A Bayesian estimator with a static prior over the dynamics sufficiently explained participants' beliefs in our data, while results were comparable with a dynamic prior that was updated at every trial. This could be attributed to the structure of the random walk of the control dynamics across trials, as a static prior is not as computationally demanding and potentially more suitable for fast changes in the time constant. These Bayesian models attempt to explain behavior in an optimal way given the task structure. Meanwhile, alternative suboptimal models (fixed estimate, carry-over estimate, sensory feedback model) failed to explain behavior successfully, especially when optic flow was available. These results strongly favor underlying computations within the context of optimality in the presence of optic flow.

Task performance was substantially worse in the vestibular condition, in a manner suggesting that vestibular inputs from motion cueing lack the reliability to precisely estimate control dynamics on individual trials. Nevertheless, the vestibular system could still facilitate inference by integrating trial history to build expectations about their statistics. Consistent with this, the mean of the prior distribution over the dynamics fit to data was very close to the mean of the true sampled distribution, suggesting that even if within-trial vestibular observations are not sufficient, participants possibly combine information about the dynamics across trials to construct their prior beliefs. This is consistent with the findings of *Prsa et al., 2015*, where vestibular cues were used to infer an underlying pattern of magnitude of motion across trials. However, the measurement of the dynamics in that study substantially differs from ours: here, motion dynamics are inferred using self-motion cues within each trial whereas in *Prsa et al., 2015*, the dynamics were inferred by integrating observations about the magnitude of the displacement across trials. If vestibular cues can in fact support inference of dynamics – as recent findings suggest in eye-head gaze shifts (*Sağlam et al., 2014*) – a common processing mechanism could be shared across sensory modalities. Overall, this finding highlights the importance of incorporating estimates of the control dynamics in models of self-motion perception and path integration.

## Limitations and future directions

Note that restrictions of our motion platform limited the range of velocities that could be tested, allowing only for relatively small velocities (see Materials and methods). Consequently, trial durations were long, but the motion platform also restricted total displacement, so we could not test larger target distances. We previously studied visual path integration with larger velocities and our results in the visual and combined conditions are comparable for similar travel times (as trials exceeded durations of 10 s, undershooting became more prevalent; *Lakshminarasimhan et al., 2018*). However, it is unclear how larger velocities (and accelerations) would affect participants' performance (especially under the vestibular condition) and whether the present conclusions are also representative of the regime of velocities not tested.

The design of the MC algorithm allowed us to circumvent the issues associated with the physical limitations of the platform to a large degree. This was achieved in part by exploiting the tilt/translation ambiguity and substituting linear translation with tilt (see Materials and methods). However, high-frequency accelerations, as those found at movement onset, generated tilts that briefly exceeded the tilt-detection threshold of the semicircular canals (*Figure 1—figure supplement 2*). Although the duration of suprathreshold stimulation was very small, we cannot exclude the possibility that the perceived tilt affected the interpretation of vestibular inputs. For example, participants may not attribute tilt to linear translation, hence underestimating their displacement. This, however, would lead to overshooting to compensate for the lack of perceived displacement,which is not what we observed in our experiment. Another potential explanation for the poor vestibular performance could be that participants perceive tilt as a conflicting cue with respect to their expected motion or visual cues. In that case, participants would only use the vestibular inputs to a small extent if at all. Manipulating vestibular inputs (e.g. gain, noise manipulations) in future experiments, either alone or in conjunction with visual cues, would offer valuable insights on two fronts: first, to help clarify the efficiency of our MC algorithm and its implications on the design of driving simulators in the future, and second, to precisely quantify the contribution of vestibular cues to path integration in natural settings.

For the sake of simplicity, we modeled each trial's control dynamics as a single measurement per trial when, in reality, participants must infer the dynamics over the course of a trial using a dynamic process of evidence accumulation. Specifically, participants must measure their self-motion velocity over time and combine a series of measurements to extract information about the underlying dynamics. Although we were able to explain the experimental findings of the influence of control dynamics on steering responses with our model, this approach could be expanded into a more normative framework using hierarchical Bayesian models (*Mathys et al., 2011*) to infer subjective position estimates by marginalizing over possible control dynamics.

One interesting question is whether providing feedback would eliminate the inference bias of the control dynamics estimation and future studies should explicitly test this hypothesis. Furthermore, it would be interesting to jointly introduce sensory conflict and manipulate sensory reliability to study dynamic multisensory integration such that sensory contributions during navigation can be better disentangled. Although it has been shown that cue combination takes place during path integration (*Tcheang et al., 2011*), previous studies have had contradicting results regarding the manner in which body-based and visual cues are combined (*Campos et al., 2010*; *Chrastil et al., 2019*; *Koppen et al., 2019*; *Petrini et al., 2016*; *ter Horst et al., 2015*). Since visual and vestibular signals differ in their sensitivity to different types of motion (*Karmali et al., 2014*), the outcomes of their integration may depend on the self-motion stimuli employed. Combined with hierarchical models of self-motion inference that considers the control dynamics, it is possible to develop an integrated, multi-level model of navigation, while constraining dramatically the hypothesized brain computations and their neurophysiological correlates.

## Materials and methods
### Equipment and task
Fifteen participants (9 male, 6 female; all adults in the age group 18–32) participated in the experiments. Apart from two participants, all participants were unaware of the purpose of the study. Experiments were first performed in the above two participants before testing others. All experimental procedures were approved by the Institutional Review Board at Baylor College of Medicine and all participants signed an approved consent form.

### Experimental setup
The participants sat comfortably on a chair mounted on an electric motor allowing unrestricted yaw rotation (Kollmorgen motor DH142M-13–1320, Kollmorgen, Radford, VA), itself mounted on a six-degree-of-freedom motion platform (comprised of MOOG 6DOF2000E, Moog Inc, East Aurora, NY). Participants used an analog joystick (M20U9T-N82, CTI electronics, Stratford, CT) with two degrees of freedom and a circular displacement boundary to control their linear and angular speed in a virtual environment based on visual and/or vestibular feedback. The visual stimulus was projected (Canon LV-8235 UST Multimedia Projector, Canon USA, Melville, NY) onto a large rectangular screen (width ×

height : 158 × 94 cm) positioned in front of the participant (77 cm from the rear of the head). Participants wore crosstalk free ferroelectric active-shutter 3D goggles (RealD CE4s, ColorLink Japan, Ltd, Tokyo, Japan) to view the stimulus. Participants wore headphones generating white noise to mask the auditory motion cues. The participants' head was fixed on the chair using an adjustable CIVCO FirmFit Thermoplastic face mask (CIVCO, Coralville, IA).

Spike2 software (Power 1401 MkII data acquisition system from Cambridge Electronic Design Ltd, Cambridge, United Kingdom) was used to record joystick and all event markers for offline analysis at a sampling rate of $833\frac{1}{3}$ Hz.

## Visual stimulus

Visual stimuli were generated and rendered using C++ Open Graphics Library (OpenGL) by continuously repositioning the camera based on joystick inputs to update the visual scene at 60 Hz. The camera was positioned at a height of 70 cm above the ground plane, whose textural elements lifetimes were limited (~250 ms) to avoid serving as landmarks. The ground plane was circular with a radius of 37.5 m (near and far clipping planes at 5 and 3750 cm, respectively), with the participant positioned at its center at the beginning of each trial. Each texture element was an isosceles triangle (base × height 5.95 × 12.95 cm) that was randomly repositioned and reoriented at the end of its lifetime. The floor density was held constant across trials at $\rho = 2.5$ elements/m$^2$. The target, a circle of radius 25 cm whose luminance was matched to the texture elements, flickered at 5 Hz and appeared at a random location between $\theta = \pm 38\ deg$ of visual angle at a distance of $r = 2.5 - 5.5$ m (average distance $\bar{r} = 4$ m) relative to where the participant was stationed at the beginning of the trial. The stereoscopic visual stimulus was rendered in an alternate frame sequencing format and participants wore active-shutter 3D goggles to view the stimulus.

## Behavioral task – visual, inertial, and multisensory motion cues

Participants were asked to navigate to a remembered target ('firefly') location on a horizontal virtual plane using a joystick, rendered in 3D from a forward-facing vantage point above the plane. Participants pressed a button on the joystick to initiate each trial and were tasked with steering to a randomly placed target that was cued briefly at the beginning of the trial. A short tone at every button push indicated the beginning of the trial and the appearance of the target. After 1 s, the target disappeared, which was a cue for the participant to start steering. During steering, visual and/or vestibular/somatosensory sensory feedback was provided (see below). Participants were instructed to stop at the remembered target location, and then push the button to register their final position and start the next trial. Participants did not receive any feedback about their performance. Prior to the first session, all participants performed about 10 practice trials to familiarize themselves with joystick movements and the task structure.

The three sensory conditions (visual, vestibular, combined) were randomly interleaved. In the visual condition, participants had to navigate toward the remembered target position given only visual information (optic flow). Visual feedback was stereoscopic, composed of flashing triangles to provide self-motion information but no landmark. In the vestibular condition, after the target disappeared, the entire visual stimulus was shut off too, leaving the participants to navigate in complete darkness using only vestibular/somatosensory cues generated by the motion platform. In the combined condition, participants were provided with both visual and vestibular information during their movement.

Independently of the manipulation of the sensory information, the properties of the motion controller also varied from trial to trial. Participants experienced different time constants in each trial, which affected the type and amount of control that was required to complete the task. In trials with short time constants, joystick position mainly controlled velocity, whereas in trials with long time constants, joystick position approximately controlled the acceleration (explained in detail in *Control dynamics* below).

Each participant performed a total of about 1450 trials (mean ± standard deviation (SD): 1450 ± 224), split equally among the three sensory conditions (mean ± SD – vestibular: 476 ± 71, visual: 487 ± 77, combined: 487 ± 77). We aimed for at least 1200 total trials per participant, and collected extended data from participants whose availability was compatible with the long runtime of our experiment.

## Joystick control

Participants navigated in the virtual environment using a joystick placed in front of the participant's midline, in a holder mounted on the bottom of the screen. This ensured that the joystick was parallel to the participant's vertical axis, and its horizontal orientation aligned to the forward movement axis. The joystick had two degrees of freedom that controlled linear and angular motion. Joystick displacements were physically bounded to lie within a disk, and digitally bounded to lie within a square. Displacement of the joystick over the anterior-posterior axis resulted in forward or backward translational motion, whereas displacement in the left-right axis resulted in rotational motion. The joystick was enabled after the disappearance of the target. To avoid skipping trials and abrupt stops, the button used to initiate trials was activated only when the participant's velocity dropped below 1 cm/s.

The joystick controlled both the visual and vestibular stimuli through an algorithm that involved two processes. The first varied the control dynamics, producing velocities given by a lowpass filtering of the joystick input, mimicking an inertial body under viscous damping. The time constant for the control filter (control timescale) was varied from trial to trial, according to a correlated random process as explained below.

The second process was an MC algorithm applied to the output of the control dynamics process, which defined physical motion that approximated the accelerations an observer would feel under the desired control dynamics, while avoiding the hardwired constraints of the motion platform. This MC algorithm trades translation for tilt, allowing extended acceleration without hitting the displacement limits (24 cm).

These two processes are explained in detail below.

## Control dynamics

Inertia under viscous damping was introduced by applying a lowpass filter on the control input, following an exponential weighted moving average with a time constant that slowly varied across trials. On each trial, the system state evolved according to a first-order Markov process in discrete time, such that the movement velocity at the next time step depended only on the current joystick input and the current velocity. Specifically, the vertical and horizontal joystick positions $u_t^{v}$ and $u_t^{\omega}$ determined the linear and angular velocities $v_t$ and $\omega_t$ as:

$$v_{t+1} = \alpha v_t + \beta_v u_t^{v} \text{ and } \omega_{t+1} = \alpha \omega_t + \beta_\omega u_t^{\omega} \tag{1a}$$

The time constant $\tau$ of the lowpass filter determined the coefficient $\alpha$ (**Figure 1—figure supplement 3A**):

$$\alpha = e^{\frac{-\Delta t}{\tau}} \tag{1b}$$

Sustained maximal controller inputs of $u_t^{v} = 1$ or $u_\omega^{v} = 1$ produce velocities that saturate at:

$$\nu_{\max} = \beta_\nu/(1-\alpha) \quad \text{and} \quad \omega_{\max} = \beta_\omega/(1-\alpha) \tag{1c}$$

We wanted to set $v_{\max}$ and $\omega_{\max}$ in such a way that would ensure that a target at an average linear or angular displacement $x$ is reachable in an average time $T$, regardless of $\tau$ (we set $x = 4$ m and $T = 8.5$ s). This constrains the input gains $\beta_v$ and $\beta_\omega$. We derived these desired gains based on a 1D bang-bang control model (i.e. purely forward movement, or pure turning) which assumes maximal positive control until time $s$, followed by maximal negative control until time $T$ (**Figure 1—figure supplement 3A**). Although we implemented the leaky integration in discrete time with a frame rate of 60 Hz, we derived the input gains using continuous time and translated them to discrete time.

The velocity at any time $0 \leq t \leq T$ during the control is:

$$\frac{v_t}{v_{\max}} = \begin{cases} 1 - e^{\frac{-t}{\tau}} & 0 < t \leq s \\ -1 + \left(\frac{v_s}{v_{\max}} + 1\right) e^{-\frac{t-s}{\tau}} & s < t < T \end{cases} \tag{1d}$$

where $v_s$ is the velocity at the switching time $s$ when control switched from positive to negative, given by:

$$v_s = v_{\max} \left(1 - e^{\frac{-s}{\tau}}\right) \tag{1e}$$

By substituting $v_s$ into **Equation 1d** and using the fact that at time $T$, the controlled velocity should return to 0, we obtain an expression that we can use to solve for $s$:

$$v_T = 0 = -1 + \left( \frac{v_{max}\left(1 - e^{\frac{-s}{\tau}}\right)}{v_{max}} + 1 \right) e^{-\frac{T-s}{\tau}} \tag{1f}$$

Observe that $v_{max}$ cancels in this equation, so the switching time $s$ is independent of $v_{max}$ and therefore also independent of the displacement $x$ (see also **Figure 1—figure supplement 3A**):

$$s = \tau \log \left( \frac{1 + e^{\frac{T}{\tau}}}{2} \right) \tag{1g}$$

Integrating the velocity profile of **Equation 1d** to obtain the distance travelled by time $T$, substituting the switch time $s$ (**Figure 1—figure supplement 3A**), and simplifying, we obtain:

$$x = x_T = 2\,\tau\,v_{max}\log\left(\cosh\frac{T}{2\tau}\right) \tag{1h}$$

We can then solve for the desired maximum linear speed $v_{max}$ for any time constant $\tau$, average displacement $x$, and trial duration $T$:

$$v_{max}\left(\tau\right) = \frac{x}{2\tau}\frac{1}{\log\cosh\left(T/2\tau\right)} \tag{1i}$$

Similarly, the maximum angular velocity was: $\omega_{max}\left(\tau\right) = \frac{\theta}{2\tau}\frac{1}{\log\cosh\left(T/2\tau\right)}$ , where $\theta$ is the average angle we want our participant to be able to turn within the average time $T$.

These equations can also be re-written in terms of a dimensionless time $z = \tau/T$ (duration of trial in units of the time constant) and average velocities $\bar{v} = x/T$ and $\bar{\omega} = \theta/T$:

$$v_{max} = \bar{v}\frac{1/2z}{\log\cosh\left(1/2z\right)} \quad \text{and} \quad \omega_{max} = \bar{\omega}\frac{1/2z}{\log\cosh\left(1/2z\right)} \tag{1j}$$

where $\theta$ is the average angle we want the participants to be able to steer within time $T$.

Setting control gains according to **Equation 1i** allows us to manipulate the control timescale $\tau$, while approximately maintaining the average trial duration for each target location (**Figure 1—figure supplement 3B**). Converting these maximal velocities into discrete-time control gains using **Equations 1.1–1.3** gives us the desired inertial control dynamics.

## Slow changes in time constant

The time constant $\tau$ was sampled according to a temporally correlated log-normal distribution. The log of the time constant, $\phi = \log\tau$ , followed a bounded random walk across trials according to (**Figure 1—figure supplement 3C**)

$$\phi_{t+1} = c\,\phi_t + \eta_t \tag{2}$$

The marginal distribution of $\phi$ was normal, $\mathcal{N}\left(\mu_\phi, \sigma_\phi^2\right)$ , with mean $\mu_\phi = \frac{1}{2}\left(\ln\tau_- + \ln\tau_+\right)$ and standard deviation $\sigma_\phi = \frac{1}{4}\left(\log\tau_+ - \log\tau_-\right)$ , which ensured that 95% of the velocity timescales lay between $\tau_-$ and $\tau_+$ . The velocity timescales changed across trials with their own timescale $\tau_\phi$ , related to the update coefficient by $c = e^{\frac{-\Delta t}{\tau_\phi}}$ , where we set $\Delta t$ to be one trial and $\tau_\phi$ to be two trials. To produce the desired equilibrium distribution of $\phi$ we set the scale of the random walk Gaussian noise $\eta \sim \mathcal{N}\left(\mu_\eta, \sigma_\eta^2\right)$ with $\mu_\eta = \mu_\phi\left(1 - c\right)$ and $\sigma_\eta^2 = \sigma_\phi^2\left(1 - c^2\right)$ .

## MC algorithm

Each motion trajectory consisted of a linear displacement in the 2D virtual space combined with a rotation in the horizontal plane. While the motion platform could reproduce the rotational movement using the yaw motor (which was unconstrained in movement range and powerful enough to render any angular acceleration or speed in this study), its ability to reproduce linear movement was limited by the platform's maximum range of 25 cm and maximum velocity of 50 cm/s (in practice, the platform was powerful enough to render any linear acceleration in this study). To circumvent this limitation, we

designed an MC algorithm that takes advantage of the gravito-inertial ambiguity (**Einstein, 1907**) inherent to the vestibular organs (**Angelaki and Dickman, 2000**; **Fernandez et al., 1972**; **Fernández and Goldberg, 1976**).

Specifically, the otolith organs in the inner ear sense both linear acceleration (*A*) and gravity (*G*), that is, they sense the gravito-inertial acceleration (GIA): $F = G + A$. Consequently, a forward acceleration of the head ($a_x$, expressed in *g*, with 1*g* = 9.81 m/s²) and a backward pitch (by an angle $\theta$, in radians) will generate a total GIA $Fx = \theta + a_x$. The MC took advantage of this ambiguity to replace linear acceleration by tilt. Specifically, it controlled the motion platform to produce a total GIA (**Figure 1—figure supplement 1**, 'desired platform GIA') that matched the linear acceleration of the simulated motion in the virtual environment. As long as the rotation that induced this simulated acceleration was slow enough, the motion felt subjectively was a linear acceleration.

This control algorithm was based on a trade-off where the high-pass component of the simulated inertial acceleration (**Figure 1—figure supplement 1**, 'desired platform linear acceleration') was produced by translating the platform, whereas the low-pass component was produced by tilting the platform (**Figure 1—figure supplement 1**, 'desired platform tilt').

Even though this method is generally sufficient to ensure that platform motion remains within its envelope, it does not guarantee it. Thus, the platform's position, velocity, and acceleration commands were fed through a sigmoid function $f$ (**Figure 1—figure supplement 1**, 'platform limits'). This function was equal to the identity function ($f(x) = x$) as long as motion commands were within 75% of the platform's limits, so these motion commands were unaffected. When motion commands exceed this range, the function bends smoothly to saturate at a value set slightly below the limit, thus preventing the platform from reaching its mechanical range (in position, velocity, or acceleration) while ensuring a smooth trajectory. Thus, if the desired motion exceeds 75% of the platform's performance envelope, the actual motion of the platform is diminished, such that the total GIA actually experienced by the participant ('actual platform GIA') may not match the desired GIA. If left uncorrected, these GIA errors would result in a mismatch between inertial motion and the visual VR stimulus. To prevent these mismatches, we designed a loop that estimates GIA error and updates the simulated motion in the visual environment. For instance, if the joystick input commands a large forward acceleration and the platform is unable to reproduce this acceleration, then the visual motion is updated to represent a slower acceleration that matches the platform's motion. Altogether, the IC and MC algorithms are applied sequentially as follows: (1) The velocity signal produced by the IC process controls the participant's attempted motion in the virtual environment. (2) The participant acceleration in the VR environment is calculated and inputted to the MC algorithm ('desired platform GIA'). (3) The MC computes the platform's motion commands and the actual platform GIA is computed. (4) The difference between the desired GIA motion actual GIA (GIA error) is computed and used to update the motion in the virtual environment. (5) The updated position is sent to the visual display.

A summary of the performance and efficiency of the MC algorithm during the experiment can be seen in **Figure 1—figure supplement 2**. For a detailed view of the implementation of the MC algorithm, refer to Appendix 1.

## Quantification and statistical analysis

Customized MATLAB code was written to analyze data and to fit models. Depending on the quantity estimated, we report statistical dispersions either using 95% confidence interval, standard deviation, or standard error in the mean. The specific dispersion measure is identified in the portion of the text accompanying the estimates. For error bars in figures, we provide this information in the caption of the corresponding figure. We report and describe the outcome as significant if p<0.05.

### Estimation of response gain

In each sensory condition, we first computed the $\tau$-independent gain for each participant; we regressed (without an intercept term) each participant's response positions ($\tilde{r}, \tilde{\theta}$) against target positions ($r, \theta$) separately for the radial ($\tilde{r}$ vs. $r$) and angular ($\tilde{\theta}$ vs. $\theta$) coordinates, and the radial and angular response gains ($g_r$, $g_\theta$) were quantified as the slope of the respective regressions (**Figure 2A**). In addition, we followed the same process to calculate gain terms within three $\tau$ groups of equal size (**Figure 3A**).

## Correlation between residual error and time constant τ

To evaluate the influence of the time constant on the steering responses, we computed the correlation coefficient between the time constants and the residual errors from the mean response (estimated using the response gain) for distance and angle. Under each sensory condition, the radial residual error ($\varepsilon_r$) for each trial   was given by:

$$\varepsilon_{r,i} = \tilde{r}_i - g_r r_i \tag{3a}$$

where $\tilde{r}_i$ is the radial response, and the mean radial response is given by multiplying the target distance $r_i$ by the radial gain $g_r$ . Similarly, the angular residual error ($\varepsilon_\theta$) was calculated as:

$$\varepsilon_{\theta,i} = \tilde{\theta}_i - g_\theta \theta_i \tag{3b}$$

## Regression model containing τ

To assess the manner in which the time constant affected the steering responses, we augmented the simple linear regression models for response gain estimation mentioned above with $\tau$-dependent terms (***Figure 3—figure supplement 2***; $\tau$ and $\tau * r$ for radial response $\tilde{r}$ , $\tau$, and $\tau * \theta$ for angular response $\tilde{\theta}$). Subsequently, we calculated the Pearson's linear partial correlations between the response positions and each of the three predictors.

## Estimation of τ-dependent gain

To quantify the extent to which the time constant modulates the response gain, we linearly regressed each participant's response positions ($\tilde{r}, \tilde{\theta}$) against target positions ($r, \theta$) and the interaction between target positions and the time constant $\tau$ according to:

$$\tilde{r} = b_r r + a_r r\tau \quad \text{and} \quad \tilde{\theta} = b_\theta \theta + a_\theta \theta\tau. \tag{4a}$$

where $b_r$, $b_\theta$ and $a_r$, $a_\theta$ are the coefficients of the target locations and the interaction terms, respectively. All quantities were first standardized by dividing them with their respective standard deviation, to avoid size effects of the different predictors. This form allows for modulation of the response gain by the time constant, which is clear when the target location is factored out:

$$\tilde{r} = r\left(b_r + a_r\tau\right) \quad \text{and} \quad \tilde{\theta} = \theta\left(b_\theta + a_\theta\tau\right). \tag{4b}$$

## Estimation of simulated no-adaptation response gains

We quantified the extent to which participants failed to adapt to the underlying control dynamics, by generating a simulated null case for no adaptation. First, we selected trials in which the time constant was close to the mean of the sampling distribution (±0.2 s). Then, we integrated the steering input of those trials with time constants from other trials (see ***Equations 1a, 1b***). This generated a set of trajectories for which the steering corresponded to a different time constant, providing us with a null case of no adaptation to the underlying dynamics. We then stratified the simulated trajectories into equal-sized groups based on the time constants (same as in ***Figure 3A***) and computed the corresponding radial and angular response gains. Note that the response gains were computed according to the target locations of the initial set of trials.

## Rationale behind modeling approach

We tested the hypothesis that the $\tau$-dependent errors in steering responses arise from participants misestimating control dynamics on individual trials. Specifically, if participants' estimate of the time constant $\tau$ differs from the actual value, then their *believed* trajectories (computed using the estimated $\tau$) would differ accordingly from the *actual* trajectories along which they travelled. believed stopping locations should land on or near the target. However, various unmeasurable fluctuations in that belief across trials should lead to variability clustered around the target location. Because participants try to stop on their believed target location, the *believed* stopping locations, subject to unmeasurable fluctuations of the belief across trials, should be distributed evenly around the participant's mean response (mean belief), after adjusting for the average response gain. This is so because, if the distribution of believed responses depended on the time constant, then that would imply that participants

willingly misestimated the control dynamics. Mathematically, the *subjective* residual errors (deviation of the believed stopping location from the mean response for a given target; see Materials and methods: *Correlation between residual error and time constant τ*) should be distributed evenly around zero and be uncorrelated with the time constant $τ$. Therefore, a good model of the participants' beliefs should predict that subjective residual errors are statistically independent of the time constant.

## Bayesian observer model for *τ* estimation

To account for the effect of the time constant $τ$ on steering performance, we considered a two-step observer model that uses a measurement $m$ of the real time constant $τ$ and a prior distribution over hypothesized time constants in logarithmic scale to compute an estimate $\hat{τ}$ on each trial (first step), and then integrates the actual joystick input using that estimate to reconstruct the participant's believed trajectory (second step). We formulated our model in the logarithmic space of $φ = \log τ$, therefore the prior distribution over the hypothesized time constants $p(φ)$ was assumed to be normal in log-space with mean, $μ_{prior}$ and standard deviation, $σ_{prior}$. The measurement distribution $p(m|φ)$ was also assumed to be normal in log-space with mean $φ$, and standard deviation $σ_{likelihood}$. Note that whereas the prior $p(φ)$ remains fixed across trials of a particular sensory modality, the mean of measurement distribution is governed by $φ$ and thus varies across trials. For each sensory modality, we fit two parameters, $Θ \ni \{μ_{prior}, λ\}$, where $λ$ was taken to be the ratio of $σ_{prior}$ over $σ_{likelihood}$ (i.e. their relative weight).

## Model fitting

When inferring the participant's beliefs about the control dynamics, we computed the posterior distribution on trial $i$ as $p(φ|m_i) \propto p(φ) p(m_i|φ)$ (*Figure 5A*, left) and then selected the median over $φ$ (equal to the maximum a posteriori estimate), and back-transformed it from log-space to obtain an estimate of the time constant $\hat{τ}_i$ for that trial:

$$\hat{τ}_i = \exp\left\{\underset{φ}{\operatorname{argmax}} \ p(φ|m_i)\right\} \tag{5}$$

Subsequently, $\hat{τ}$ is used to integrate the actual joystick input and produce the participant's believed trajectory, according to (*Equations 1–10*) in the *Control dynamics* section.

The Bayesian model had two free parameters $Θ \ni \{μ_{prior}, λ\}$. We fit the model by assuming that participants stop as close to the target as possible given their understanding of the task. Specifically, we minimized the mean squared error (MSE) between the measured mean stopping position (computed using the response gains $g_r$ and $g_θ$ from *Equation 3*) and our model of the participant's believed stopping location $\mathbf{x}_i$ given the inferred dynamics $τ_i$. For each sensory condition:

$$Θ^* = \underset{Θ}{\operatorname{argmin}} \ \frac{1}{n} \sum_{i=1}^{n} \left\{\hat{\mathbf{x}}_i\left(\hat{τ}_i, \mathbf{u}_i\right) - \mathbf{G}\mathbf{x}_i^{tar}\right\}^2 \tag{6}$$

where, for each trial $i$, $\hat{\mathbf{x}}_i$ is the believed participant's position, $\hat{τ}_i$ is the estimated time constant, $\mathbf{u}_i$ is the time series of the joystick control input, $\mathbf{x}_i^{tar}$ is the actual target position, $\mathbf{G}$ is the response gain matrix determined from $g_r$ and $g_θ$, and $n$ is the total number of trials.

## Model validation

To evaluate the performance of our model, we examined the correlations between the *subjective* residual error and $τ$ that are given by the model. The subjective residual error is defined as the difference between the believed (subjective) stopping location that a model produces and the mean response of the actual trajectories, adjusted for the response gain. The subjective residual errors are calculated for the radial and angular components of the response separately, according to *Equation 3* (where actual responses $\tilde{r}, \tilde{θ}$ are substituted by believed responses $\hat{\tilde{r}}, \hat{\tilde{θ}}$, respectively). Ideally, these correlations should not exist for the model predictions (explained in text; *Figure 5B*). We determined the statistical significance of the model-implied correlations by adjusting for multiple comparisons (required level of statistical significance: p = 0.0085). To assess the performance of the Bayesian model, we compared the correlations between believed and actual stopping location with the time constant (*Figure 5B*; Wilcoxon signed-rank test).

## Dynamic prior model

Since the time constant changes randomly across trials, we tested whether the history of time constants influenced the estimate $\tau$. If true, the Bayesian model would imply a prior distribution over $\varphi = \log \tau$ that is dynamically changing according to the recent history of time constants, rather than being fixed. To explore this possibility, we repeated the two-step model outlined above, with the difference that the mean of the prior distribution is updated at every trial by a weighted average of the mean prior in the previous trial and the current measurement over $\varphi$:

$$\mu_{\text{prior}, i} = \left(1 - k\right) \mu_{\text{prior}, i-1} + k \, \varphi_i, \text{where } k = \frac{\lambda^2}{\lambda^2 + 1} \tag{7}$$

and where $\lambda$ is the ratio of prior standard deviation over likelihood standard deviation. As $k$ indicates, the relative weighting between prior and measurement on each trial depends solely on their relative widths. Finally, the initial prior was taken to be the time constant on the first trial. Thus, the only free parameter we fit was $\lambda$.

## Sensory-independent model

As another alternative to the Bayesian model with a static prior, we also constructed a model where participants ignored changes in the time constants and navigated according to a fixed estimate $\tau$ across all trials in each sensory condition. This model had only one free parameter: the time constant estimate $\tau$, which was integrated with the actual joystick input of each trial to reconstruct the believed trajectory of the participant. We fit $\tau$ for each sensory condition by minimizing the MSE between the believed stopping location and the mean response (according to *Equation 6*).

## Model comparison

To compare the static prior Bayesian model against the dynamic prior Bayesian and the sensory-independent models, we compared the correlations between believed stopping locations and time constants that each model produces (*Figure 7*; paired Student's *t*-test).

## Sensory feedback control model

We tested a sensory feedback control model, in which the controller uses bang-bang control and switches from forward to backward input at a constant and predetermined distance from the target position (corrected for the bias, i.e. mean response). Specifically, we preserved the actual angular and only fitted the linear control input for each trial. Thus, as switch distance, we refer to a Euclidean distance from the bias-corrected target position. We fit the mean and standard deviation of the switch distance for each participant in each condition separately, by minimizing the distance of the actual from the model-predicted stopping locations. To evaluate how well this model describes our data, we compared the correlations and regression slopes between the time constant and residual errors from the stopping locations predicted by the model with those from our actual data (*Figure 7—figure supplement 2*).

## Acknowledgements

We thank Jing Lin and Jian Chen for their technical support, and Baptiste Caziot, Panos Alefantis, Babis Stavropoulos, Evangelia Pappou, and Emmanouela Pechynaki for their useful insights. This work was supported by the Simons Collaboration on the Global Brain, grant no. 324143, and NIH DC007620. GCD was supported by NIH EY016178.

## Additional information

### Funding

| Funder | Grant reference number | Author |
| --- | --- | --- |
| NIH Blueprint for Neuroscience Research | NIH DC007620 | Dora E Angelaki |

| Funder | Grant reference number | Author |
|---|---|---|
| National Science Foundation | NeuroNex DBI-1707398 | Kaushik J Lakshminarasimhan |
| Gatsby Charitable Foundation | | Kaushik J Lakshminarasimhan |
| Simons Foundation | 324143 | Xaq Pitkow Dora E Angelaki |
| National Institutes of Health | R01NS120407 | Xaq Pitkow Dora E Angelaki |
| National Science Foundation | 1707400 | Xaq Pitkow |
| National Science Foundation | 1552868 | Xaq Pitkow |

The funders had no role in study design, data collection and interpretation, or the decision to submit the work for publication.

### Author contributions

Akis Stavropoulos, Conceptualization, Data curation, Formal analysis, Investigation, Methodology, Software, Validation, Visualization, Writing – original draft; Kaushik J Lakshminarasimhan, Formal analysis, Investigation, Methodology, Project administration, Software, Supervision, Validation, Visualization, Writing – original draft, Writing – review and editing; Jean Laurens, Conceptualization, Investigation, Methodology, Software, Writing – review and editing; Xaq Pitkow, Conceptualization, Funding acquisition, Investigation, Methodology, Project administration, Resources, Supervision, Validation, Writing – review and editing; Dora E Angelaki, Conceptualization, Funding acquisition, Investigation, Project administration, Resources, Validation, Writing – review and editing

### Author ORCIDs

Akis Stavropoulos ORCID http://orcid.org/0000-0003-4613-9422
Kaushik J Lakshminarasimhan ORCID http://orcid.org/0000-0003-3932-2616
Jean Laurens ORCID http://orcid.org/0000-0002-9101-2802
Xaq Pitkow ORCID http://orcid.org/0000-0001-6376-329X
Dora E Angelaki ORCID http://orcid.org/0000-0002-9650-8962

### Ethics

Human subjects: All experimental procedures were approved by the Institutional Review Board at Baylor College of Medicine and all participants signed an approved consent form (H-29411).

### Decision letter and Author response

Decision letter https://doi.org/10.7554/eLife.63405.sa1
Author response https://doi.org/10.7554/eLife.63405.sa2

## Additional files

### Supplementary files
• Transparent reporting form

### Data availability

MATLAB code implementing all quantitative analyses in this study is available online (https://github.com/AkisStavropoulos/matlab_code, copy archived at swh:1:rev:03fcaf8a8170d99f80e2bd9b-40c888df34513f89). The dataset was made available online at the following address: https://gin.g-node.org/akis_stavropoulos/humans_control_dynamics_sensory_modality_steering.

The following dataset was generated:

| Author(s) | Year | Dataset title | Dataset URL | Database and Identifier |
|---|---|---|---|---|
| Stavropoulos L, Laurens, Pitkow A | 2021 | Steering data from a navigation study in virtual reality under different sensory modalities and control dynamics, named "Influence of sensory modality and control dynamics on human path integration" | https://gin.g-node.org/akis_stavropoulos/humans_control_dynamics_sensory_modality_steering | G-node, humans_control_dynamics_sensory_modality_steering |

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

# Appendix 1

## Implementation of MC algorithm

### Step 1

In the first step, the participant's velocity is being transformed into the VR (screen) coordinates. This transformation is necessary to deduce centrifugal components from the participants' trajectory, and include them in the motor commands:

$$v_{t+1}^{\text{VR},x} = v_{t+1}^{\text{JS}} \cdot \cos\left(\varphi_t^{\text{VR}}\right)$$

$$v_{t+1}^{\text{VR},y} = v_{t+1}^{\text{JS}} \cdot \sin\left(\varphi_t^{\text{VR}}\right)$$

$$\omega_{t+1}^{\text{VR}} = \omega_{t+1}^{\text{JS}}$$

where $v^{\text{VR},x}$ and $v^{\text{VR},y}$ are the linear velocities of the participant in VR coordinates, $\omega^{\text{VR}}$ is the angular velocity of the VR system, and $\varphi^{\text{VR}}$ is the direction of the platform in space.

### Step 2

As mentioned before, the arena diameter is finite, and it is necessary to keep track of the participant's position in the arena, to avoid 'crashing' on the invisible walls. In this step, the participant's velocity is slowed down when the participant approaches the boundaries of the arena, to account for a 'smooth crash'.

### Step 3

Here, the current acceleration is calculated in the VR coordinates $\left(a^{\text{VR},x}, a^{\text{VR},y}\right)$. This is also where the GIA error feedback loop (see Step 10) updates the VR acceleration.

$$\alpha_{t+1}^{\text{VR},x} = \frac{v_{t+1}^{\text{VR},x} - v_t^{\text{VR},x} + \frac{dt}{\tau_{\text{MC}}} \cdot \left(v_t^{\text{VR},x} - \hat{v}_t^{\text{VR},x}\right)}{dt}$$

$$\alpha_{t+1}^{\text{VR},y} = \frac{v_{t+1}^{\text{VR},y} - v_t^{\text{VR},y} + \frac{dt}{\tau_{\text{MC}}} \cdot \left(v_t^{\text{VR},y} - \hat{v}_t^{\text{VR},y}\right)}{dt}$$

where $v_t$ is the updated velocity from the previous timestep ($\tau_{\text{MC}}$ explained in Step 10). After the acceleration is obtained, it is being transformed back to the participant's coordinates $\left(a^{\text{sub},x}, a^{\text{sub},y}\right)$:

$$\alpha_{t+1}^{\text{sub},x} = \alpha_{t+1}^{\text{VR},x} \cdot \cos\left(\varphi_t^{\text{VR}}\right) + \alpha_{t+1}^{\text{VR},y} \cdot \sin\left(\varphi_t^{\text{VR}}\right)$$

$$\alpha_{t+1}^{\text{sub},y} = -\alpha_{t+1}^{\text{VR},x} \cdot \sin\left(\varphi_t^{\text{VR}}\right) + \alpha_{t+1}^{\text{VR},y} \cdot \cos\left(\varphi_t^{\text{VR}}\right)$$

### Step 4

Now, the acceleration $a^{\text{sub}}$ in participant's coordinates is being transformed into platform coordinates to take into account the orientation of the participant onto the motion platform ($\varphi_t^{\text{moog}}$), which is controlled by the yaw motor. For instance, if the participant faces toward the left of the platform and accelerates forward in egocentric coordinates, then the platform should move to the left:

$$\alpha_{t+1}^{\text{desired},x} = \alpha_{t+1}^{\text{sub},x} \cdot \cos\left(\varphi_t^{\text{moog}}\right) - \alpha_{t+1}^{\text{sub},y} \cdot \sin\left(\varphi_t^{\text{moog}}\right)$$

$$\alpha_{t+1}^{\text{desired},y} = \alpha_{t+1}^{\text{sub},x} \cdot \sin\left(\varphi_t^{\text{moog}}\right) + \alpha_{t+1}^{\text{sub},y} \cdot \cos\left(\varphi_t^{\text{moog}}\right)$$

where $\alpha_{t+1}^{\text{desired},x}$ is the desired platform acceleration.

## Step 5

This is the MC step. Here, the amount of tilt and translation that will be commanded is computed, based on the tilt-translation trade-off we set. First, the platform's desired acceleration is computed by applying a step response function $f(t)$ to the acceleration input:

$$a^{\text{MC},x}(t) = \int_0^{+\infty} a^{\text{desired},x}(t) \cdot f(t-s) \ ds$$

where:

$$f(t) = k_1 \cdot e^{\frac{-t}{T_1}} + k_2 \cdot e^{\frac{-t}{T_2}} + k_3 \cdot e^{\frac{-t}{T_3}}$$

$$T = [0.07 \ \ 0.3 \ \ 1], K = [-0.4254 \ \ 1.9938 \ \ -0.5684]$$

These coefficients were adjusted with respect to the following constraints:

1. $f(0) = 1$, that is, the output would correspond to the input at $t = 0$. This was chosen to ensure that the high-frequency content of the motion would be rendered by translating the platform.
2. $\int_0^\infty f = 0$: This was chosen to ensure that, if the input was an infinitely long acceleration, the motion of the platform would stabilize to a point where the linear velocity was 0.
3. $df/dt = 0$ at $t = 0$. This was chosen because tilt velocity of the platform is equal to $-df/dt$. Since the tilt velocity at $t < 0$ is zero, this constraint ensures that tilt velocity is continuous and prevents excessive angular acceleration at $t = 0$.

The same process is repeated for the $y$ component of the acceleration.

Finally, the amount of tilt ($\theta$, in degrees) is calculated based on the difference between the desired platform motion and the deliverable motion:

$$\theta_{t+1}^{\text{MC},x} = \sin^{-1}\left(\frac{a_{t+1}^{\text{moog},x} - a_{t+1}^{\text{MC},x}}{g}\right)$$

$$\theta_{t+1}^{\text{MC},y} = \sin^{-1}\left(\frac{a_{t+1}^{\text{moog},y} - a_{t+1}^{\text{MC},y}}{g}\right)$$

where $g = 9.81$ m/s$^2$.

## Step 6

Afterward, the tilt velocity and acceleration are being calculated:

$$\dot{\theta}_{t+1}^{\text{MC},y} = \frac{\theta_{t+1}^{\text{MC},y} - \theta_t^{\text{MC},y}}{dt} \ , \ \dot{\theta}_{t+1}^{\text{MC},x} = \frac{\theta_{t+1}^{\text{MC},x} - \theta_t^{\text{MC},x}}{dt} \ ,$$

$$\ddot{\theta}_{t+1}^{\text{MC},y} = \frac{\dot{\theta}_{t+1}^{\text{MC},y} - \dot{\theta}_t^{\text{MC},y}}{dt} \ , \ \ddot{\theta}_{t+1}^{\text{MC},x} = \frac{\dot{\theta}_{t+1}^{\text{MC},x} - \dot{\theta}_t^{\text{MC},x}}{dt} \ ,$$

In a next step, we compute the motion command that should be sent by the platform. Note that the platform is placed at a height $h$ below the head. Therefore, tilting the platform by an angle $\theta$ induces a linear displacement of the head corresponding to $-h \cdot \theta \cdot \frac{\pi}{180}$ . Therefore, a linear displacement is added to the platform's motion to compensate for this. Next, we limit the platform's acceleration, velocity, and position commands to ensure that they remain within the limit of the actuators. For this purpose, we define the following function $f_{\lambda,x_{\max}}(x)$:

$$\left\{ \begin{array}{cc} \text{if } |x| \leq \lambda \cdot x_{\max} \, , & f_{\lambda,x_{\max}}(x) = x \\ \text{else if } |x| \leq (2-\lambda) \cdot x_{\max} \, , & f_{\lambda,x_{\max}}(x) = x_{\max}.\text{sign}(x) . \left[ \left| x/x_{\max} \right| - \frac{1}{4.(1-\lambda)} \cdot \left( |x/x_{\max}| - \lambda \right)^2 \right] \\ \text{if } |x| > (2-\lambda) \cdot x_{\max} \, , & f_{\lambda,x_{\max}}(x) = \text{sign}(x) . x_{\max} \end{array} \right\}$$

This function is designed so that if the input $x$ increases continuously, for example, $x(t) = t$, then the output $f_{\lambda,x_{\max}}(x(t))$ will be identical to $x$ until $x$ reaches a threshold $\lambda \cdot x_{\max}$ . After this, the

output will decelerate continuously ($\frac{d f_{\lambda,x_{\max}}(x(t))}{dt}$ = constant) until it stops at a value $x_{\max}$. We fed the platform's acceleration, velocity, and position command through this function, as follows:

$$a_{t+1}^{\text{moog},x} = f_{\lambda,a_{\max}}\left(a_{t+1}^{\text{MC},x} + h \cdot \theta_{t+1}^{\text{MC},x} \cdot \frac{\pi}{180}\right)$$

$$v_{t+1}^{\text{moog},x} = f_{\lambda,v_{\max}}\left(v_t^{\text{moog},x} + dt.a_{t+1}^{\text{moog},x}\right)$$

$$x_{t+1}^{\text{moog},x} = f_{\lambda,x_{\max}}\left(x_t^{\text{moog},x} + dt.x_{t+1}^{\text{moog},x}\right)$$

The same operation takes place for the $y$ component of the acceleration, as well as for the platform velocity and position. The process is repeated for the tilt command itself.

We set $\lambda = 0.75$ and $a_{\max} = 4$ m/s$^2$ , $v_{\max} = 0.4$ m/s, $x_{\max} = 0.23$ m, $\theta_{\max} = 300$ deg/s, $\dot{\theta}_{\max} = 30$ deg/s and $\theta_{max} = 10$ deg, slightly below the platform's and actuator physical limits. This ensured that the platform's motion matched exactly the MC algorithm's output, as long as it stayed within 75% of the platform's range. Otherwise, the function $f$ ensured a smooth trajectory and, as detailed in Steps 8–10, a feedback mechanism was used to update the participant position in the VR environment, so as to guarantee that visual motion always matched inertial motion.

## Step 7
The motor commands for tilt and translation are being sent to the platform:

$$\left[x_{t+1}^{\text{moog}}, \ y_{t+1}^{\text{moog}}, \ \theta_{t+1}^{\text{moog},x}, \ \theta_{t+1}^{\text{moog},y}\right]$$

## Step 8
Because of Step 6, the total GIA of the platform may differ from what is commanded by the MC algorithm. To detect and discrepancy, we computed the GIA provided by the platform:

$$v_{t+1}^{\text{actual},x} = \frac{\left(x_{t+1}^{\text{moog}} - x_t^{\text{moog}}\right)}{dt} \ , \ v_{t+1}^{\text{actual},y} = \frac{\left(y_{t+1}^{\text{moog}} - y_t^{\text{moog}}\right)}{dt}$$

$$a_{t+1}^{\text{actual},x} = \frac{\left(v_{t+1}^{\text{actual},x} - v_t^{\text{actual},x}\right)}{dt} \ , \ a_{t+1}^{\text{actual},y} = \frac{\left(v_{t+1}^{\text{actual},y} - y_v^{\text{actual},y}\right)}{dt}$$

$$GIA_{t+1}^{\text{actual},x} = a_{t+1}^{\text{actual},x} + g \cdot \sin\theta_{t+1}^{\text{moog},x} - h \cdot \ddot{\theta}_{t+1}^{\text{moog},x} \cdot \frac{\pi}{180}$$

$$GIA_{t+1}^{\text{actual},y} = a_{t+1}^{\text{actual},y} + g \cdot \sin\theta_{t+1}^{\text{moog},y} - h \cdot \ddot{\theta}_{t+1}^{\text{moog},y} \cdot \frac{\pi}{180}$$

## Step 9
We transform platform's GIA into participant's reference frame:

$$GIA_{t+1}^{\text{sub},x} = GIA_{t+1}^{\text{actual},x} \cdot \cos\varphi_t^{\text{moog}} + GIA_{t+1}^{\text{actual},y} \cdot \sin\varphi_t^{\text{moog}}$$

$$GIA_{t+1}^{\text{sub},y} = -GIA_{t+1}^{\text{actual},x} \cdot \sin\varphi_t^{\text{moog}} + GIA_{t+1}^{\text{actual},y} \cdot \cos\varphi_t^{\text{moog}}$$

Also, the error $e^{sub}$ between the obtained GIA and desired GIA (from Step 3) is calculated, and fed through the same sigmoid function ($\lambda = 0.75$, $GIA_{\max} = 1 \frac{\text{m}}{\text{s}^2}$) discussed previously, to avoid computational instability in the case of a big mismatch:

$$e_{t+1}^x = f_{\lambda,GIA_{\max}}\left(GIA_{t+1}^{\text{sub},x} - a_{t+1}^{\text{sub},x}\right)$$

$$e_{t+1}^y = f_{\lambda,GIA_{\max}}\left(GIA_{t+1}^{\text{sub},y} - a_{t+1}^{\text{sub},y}\right)$$

## Step 10

The GIA error is now used to update the system in the case of a mismatch. First, it is transformed into VR coordinates. Then, the velocity and position in VR coordinates are recomputed based on the joystick input and on the error signal:

$$e_{t+1}^{\text{VR},x} = e_{t+1}^{\text{sub},x} \cdot \cos \varphi_t^{\text{VR}} - e_{t+1}^{\text{sub},y} \cdot \sin \varphi_t^{\text{VR}}$$

$$e_{t+1}^{\text{VR},y} = e_{t+1}^{\text{sub},x} \cdot \sin \varphi_t^{\text{VR}} + e_{t+1}^{\text{sub},y} \cdot \cos \varphi_t^{\text{VR}}$$

$$\hat{v}_{t+1}^{\text{VR},x} = \hat{v}_t^{\text{VR},x} + \left( a_{t+1}^{\text{VR},x} + e_{t+1}^{\text{VR},x} \right) \cdot dt$$

$$\hat{v}_{t+1}^{\text{VR},y} = \hat{v}_t^{\text{VR},y} + \left( a_{t+1}^{\text{VR},y} + e_{t+1}^{\text{VR},y} \right) \cdot dt$$

$$x_{t+1}^{\text{VR},x} = x_t^{\text{VR},x} + \hat{v}_{t+1}^{\text{VR},x} \cdot dt$$

$$x_{t+1}^{\text{VR},y} = x_t^{\text{VR},y} + \hat{v}_{t+1}^{\text{VR},y} \cdot dt$$

$$\varphi_{t+1}^{\text{VR}} = \varphi_t^{\text{VR}} + \omega_t^{\text{VR}} \cdot dt$$

Note that the error signal is also fed into the acceleration in VR coordinates (see Step 3). Ideally, linear acceleration should be computed based on the updated velocity value at time $t$, that is:

$$\alpha_{t+1}^{\text{VR},x} = \frac{v_{t+1}^{\text{VR},x} - \hat{v}_t^{\text{VR},x}}{dt}$$

However, we found that this led to numerical instability, and instead we introduced a time constant $\tau_{\text{MC}} = 1\text{s}$ in the computation, as shown in Step 3.

# Appendix 2

**Appendix 2—table 1.** Average radial (top) and angular (bottom) behavioral response gains across participants, for groups of time constant $\tau$ magnitudes (mean ± SEM).

| Radial bias table | | | |
|---|---|---|---|
| | Vestibular | Visual | Combined |
| $\tau$: [0.34–1.53] | 0.649 ± 0.056 | 0.818 ± 0.057 | 0.786 ± 0.055 |
| $\tau$: [1.53–2.16] | 0.733 ± 0.063 | 0.871 ± 0.059 | 0.836 ± 0.056 |
| $\tau$: [2.16–8.89] | 0.902 ± 0.077 | 0.944 ± 0.061 | 0.917 ± 0.058 |

| Angular bias table | | | |
|---|---|---|---|
| | Vestibular | Visual | Combined |
| $\tau$: [0.34–1.53] | 0.731 ± 0.053 | 0.919 ± 0.036 | 0.902 ± 0.032 |
| $\tau$: [1.53–2.16] | 0.770 ± 0.060 | 0.984 ± 0.038 | 0.944 ± 0.029 |
| $\tau$: [2.16–8.89] | 0.878 ± 0.061 | 1.024 ± 0.040 | 1.012 ± 0.033 |

**Appendix 2—table 2.** Pearson's correlation coefficient ($r$) and corresponding p-value (p) for radial (top) and angular (bottom) correlation between residual error and the time constant $\tau$ across participants.

Mean Pearson's $r$ ± SEM: radial component – vestibular: 0.52±0.02, visual: 0.36±0.03, combined: 0.37±0.03; angular component – vestibular: 0.23±0.02, visual: 0.23±0.03, combined: 0.26±0.03.

| Radial correlations | | | |
|---|---|---|---|
| | Vestibular | Visual | Combined |
| Subject 1 | $r = 0.585, p = 4.2 \cdot 10^{-45}$ | $r = 0.502, p = 1.2 \cdot 10^{-37}$ | $r = 0.617, p = 1.0 \cdot 10^{-59}$ |
| Subject 2 | $r = 0.622, p = 5.5 \cdot 10^{-43}$ | $r = 0.338, p = 7.4 \cdot 10^{-12}$ | $r = 0.377, p = 2.9 \cdot 10^{-14}$ |
| Subject 3 | $r = 0.433, p = 3.5 \cdot 10^{-25}$ | $r = 0.280, p = 2.7 \cdot 10^{-11}$ | $r = 0.374, p = 8.7 \cdot 10^{-20}$ |
| Subject 4 | $r = 0.492, p = 9.1 \cdot 10^{-31}$ | $r = 0.494, p = 3.1 \cdot 10^{-31}$ | $r = 0.350, p = 2.1 \cdot 10^{-15}$ |
| Subject 5 | $r = 0.411, p = 4.4 \cdot 10^{-17}$ | $r = 0.314, p = 3.4 \cdot 10^{-10}$ | $r = 0.360, p = 3.7 \cdot 10^{-13}$ |
| Subject 6 | $r = 0.601, p = 2.0 \cdot 10^{-58}$ | $r = 0.233, p = 1.2 \cdot 10^{-08}$ | $r = 0.233, p = 1.2 \cdot 10^{-08}$ |
| Subject 7 | $r = 0.606, p = 1.6 \cdot 10^{-44}$ | $r = 0.522, p = 1.5 \cdot 10^{-31}$ | $r = 0.474, p = 1.1 \cdot 10^{-25}$ |
| Subject 8 | $r = 0.477, p = 9.6 \cdot 10^{-34}$ | $r = 0.255, p = 5.7 \cdot 10^{-10}$ | $r = 0.294, p = 4.6 \cdot 10^{-13}$ |
| Subject 9 | $r = 0.478, p = 1.0 \cdot 10^{-22}$ | $r = 0.517, p = 7.9 \cdot 10^{-27}$ | $r = 0.523, p = 6.3 \cdot 10^{-28}$ |
| Subject 10 | $r = 0.573, p = 7.2 \cdot 10^{-39}$ | $r = 0.497, p = 4.7 \cdot 10^{-28}$ | $r = 0.576, p = 3.5 \cdot 10^{-39}$ |
| Subject 11 | $r = 0.375, p = 5.9 \cdot 10^{-16}$ | $r = 0.224, p = 2.1 \cdot 10^{-06}$ | $r = 0.144, p = 0.002$ |
| Subject 12 | $r = 0.522, p = 2.1 \cdot 10^{-39}$ | $r = 0.341, p = 1.3 \cdot 10^{-16}$ | $r = 0.319, p = 1.1 \cdot 10^{-14}$ |
| Subject 13 | $r = 0.512, p = 1.1 \cdot 10^{-38}$ | $r = 0.385, p = 1.4 \cdot 10^{-21}$ | $r = 0.401, p = 4.7 \cdot 10^{-23}$ |
| Subject 14 | $r = 0.461, p = 8.3 \cdot 10^{-30}$ | $r = 0.241, p = 1.3 \cdot 10^{-08}$ | $r = 0.276, p = 7.0 \cdot 10^{-11}$ |
| Subject 15 | $r = 0.703, p = 1.7 \cdot 10^{-61}$ | $r = 0.214, p = 1.1 \cdot 10^{-05}$ | $r = 0.213, p = 1.3 \cdot 10^{-05}$ |

| Angular correlations | | | |
|---|---|---|---|
| | Vestibular | Visual | Combined |
| Subject 1 | $r = 0.254, p = 1.9 \cdot 10^{-08}$ | $r = 0.302, p = 1.8 \cdot 10^{-13}$ | $r = 0.437, p = 2.3 \cdot 10^{-27}$ |
| Subject 2 | $r = 0.156, p = 0.002$ | $r = 0.287, p = 8.6 \cdot 10^{-09}$ | $r = 0.270, p = 9.2 \cdot 10^{-08}$ |
| Subject 3 | $r = 0.301, p = 2.2 \cdot 10^{-12}$ | $r = 0.274, p = 7.3 \cdot 10^{-11}$ | $r = 0.351, p = 1.7 \cdot 10^{-17}$ |

*Continued on next page*

*Continued*

| | Angular correlations | | |
|---|---|---|---|
| | **Vestibular** | **Visual** | **Combined** |
| Subject 4 | $r = 0.315$, p = $1.3 \cdot 10^{-12}$ | $r = 0.299$, p = $1.7 \cdot 10^{-11}$ | $r = 0.343$, p = $8.9 \cdot 10^{-15}$ |
| Subject 5 | $r = 0.153$, p = 0.003 | $r = 0.291$, p = $6.7 \cdot 10^{-09}$ | $r = 0.387$, p = $3.8 \cdot 10^{-15}$ |
| Subject 6 | $r = 0.292$, p = $5.9 \cdot 10^{-13}$ | $r = 0.121$, p = 0.003 | $r = 0.224$, p = $4.8 \cdot 10^{-08}$ |
| Subject 7 | $r = 0.098$, p = 0.042 | $r = 0.356$, p = $2.4 \cdot 10^{-14}$ | $r = 0.275$, p = $6.0 \cdot 10^{-09}$ |
| Subject 8 | $r = 0.346$, p = $2.0 \cdot 10^{-17}$ | $r = -0.004$, p = 0.920 | $r = 0.005$, p = 0.902 |
| Subject 9 | $r = 0.093$, p = 0.071 | $r = 0.349$, p = $4.1 \cdot 10^{-12}$ | $r = 0.348$, p = $3.1 \cdot 10^{-12}$ |
| Subject 10 | $r = 0.294$, p = $4.7 \cdot 10^{-10}$ | $r = 0.336$, p = $9.6 \cdot 10^{-13}$ | $r = 0.235$, p = $9.0 \cdot 10^{-07}$ |
| Subject 11 | $r = 0.064$, p = 0.183 | $r = -0.032$, p = 0.507 | $r = 0.027$, p = 0.575 |
| Subject 12 | $r = 0.271$, p = $1.2 \cdot 10^{-10}$ | $r = 0.278$, p = $2.7 \cdot 10^{-11}$ | $r = 0.333$, p = $5.6 \cdot 10^{-16}$ |
| Subject 13 | $r = 0.238$, p = $1.2 \cdot 10^{-08}$ | $r = 0.312$, p = $2.5 \cdot 10^{-14}$ | $r = 0.255$, p = $1.0 \cdot 10^{-09}$ |
| Subject 14 | $r = 0.215$, p = $4.3 \cdot 10^{-07}$ | $r = 0.138$, p = 0.001 | $r = 0.217$, p = $3.7 \cdot 10^{-07}$ |
| Subject 15 | $r = 0.328$, p = $1.2 \cdot 10^{-11}$ | $r = 0.134$, p = 0.006 | $r = 0.137$, p = 0.005 |

**Appendix 2—table 3.** Linear regression slope coefficients for radial ($\alpha$, top) and angular ($\beta$, bottom) components of residual error against the time constant $\tau$ across participants.
Mean regression slope ± SEM: Radial (m/s) – vestibular: 0.62±0.06, visual: 0.28±0.03, combined: 0.29±0.03; angular (deg/s) – vestibular: 2.05±0.2, visual: 1.04±0.23, combined: 1.09±0.19.

| | Radial regression coefficients (m/s) | | |
|---|---|---|---|
| | **Vestibular** | **Visual** | **Combined** |
| Subject 1 | $\alpha = 0.775$ | $\alpha = 0.247$ | $\alpha = 0.337$ |
| Subject 2 | $\alpha = 0.776$ | $\alpha = 0.464$ | $\alpha = 0.470$ |
| Subject 3 | $\alpha = 0.255$ | $\alpha = 0.138$ | $\alpha = 0.157$ |
| Subject 4 | $\alpha = 0.406$ | $\alpha = 0.138$ | $\alpha = 0.269$ |
| Subject 5 | $\alpha = 1.009$ | $\alpha = 0.559$ | $\alpha = 0.487$ |
| Subject 6 | $\alpha = 0.829$ | $\alpha = 0.151$ | $\alpha = 0.149$ |
| Subject 7 | $\alpha = 0.512$ | $\alpha = 0.351$ | $\alpha = 0.330$ |
| Subject 8 | $\alpha = 0.582$ | $\alpha = 0.245$ | $\alpha = 0.222$ |
| Subject 9 | $\alpha = 0.321$ | $\alpha = 0.330$ | $\alpha = 0.311$ |
| Subject 10 | $\alpha = 0.943$ | $\alpha = 0.365$ | $\alpha = 0.445$ |
| Subject 11 | $\alpha = 0.522$ | $\alpha = 0.322$ | $\alpha = 0.177$ |
| Subject 12 | $\alpha = 0.484$ | $\alpha = 0.166$ | $\alpha = 0.210$ |
| Subject 13 | $\alpha = 0.570$ | $\alpha = 0.324$ | $\alpha = 0.327$ |
| Subject 14 | $\alpha = 0.507$ | $\alpha = 0.253$ | $\alpha = 0.321$ |
| Subject 15 | $\alpha = 0.799$ | $\alpha = 0.091$ | $\alpha = 0.102$ |

| | Angular regression coefficients (deg/s) | | |
|---|---|---|---|
| | **Vestibular** | **Visual** | **Combined** |
| Subject 1 | $\beta = 1.664$ | $\beta = 1.045$ | $\beta = 1.553$ |
| Subject 2 | $\beta = 1.645$ | $\beta = 2.022$ | $\beta = 1.632$ |
| Subject 3 | $\beta = 1.317$ | $\beta = 0.552$ | $\beta = 1.232$ |

*Continued*

| | Angular regression coefficients (deg/s) | | |
|---|---|---|---|
| | **Vestibular** | **Visual** | **Combined** |
| Subject 4 | $\beta$ = 2.165 | $\beta$ = 0.919 | $\beta$ = 1.155 |
| Subject 5 | $\beta$ = 2.349 | $\beta$ = 3.201 | $\beta$ = 3.045 |
| Subject 6 | $\beta$ = 2.620 | $\beta$ = 0.563 | $\beta$ = 0.870 |
| Subject 7 | $\beta$ = 1.434 | $\beta$ = 1.101 | $\beta$ = 0.843 |
| Subject 8 | $\beta$ = 4.185 | $\beta$ = –0.039 | $\beta$ = 0.040 |
| Subject 9 | $\beta$ = 1.254 | $\beta$ = 1.562 | $\beta$ = 1.394 |
| Subject 10 | $\beta$ = 2.937 | $\beta$ = 1.971 | $\beta$ = 1.152 |
| Subject 11 | $\beta$ = 1.849 | $\beta$ = –0.193 | $\beta$ = 0.194 |
| Subject 12 | $\beta$ = 1.382 | $\beta$ = 0.836 | $\beta$ = 0.954 |
| Subject 13 | $\beta$ = 1.619 | $\beta$ = 1.233 | $\beta$ = 1.165 |
| Subject 14 | $\beta$ = 2.141 | $\beta$ = 0.585 | $\beta$ = 0.790 |
| Subject 15 | $\beta$ = 2.214 | $\beta$ = 0.256 | $\beta$ = 0.264 |

**Appendix 2—table 4.** Partial correlation coefficients (mean ± standard deviation) for prediction of the radial ($\tilde{r}$, top) and angular ($\tilde{\theta}$, bottom) components of the final stopping location (relative to starting position) from initial target distance ($r$) and angle ($\theta$), the time constant $\tau$, and the interaction of the two ($r{\cdot}\tau$ or $r{\cdot}\theta$), respectively.

| | | Radial partial correlation coefficients ± standard deviation | | |
|---|---|---|---|---|
| | | **Vestibular** | **Visual** | **Combined** |
| | Radial distance ($r$) | 0.20 ± 0.05 | 0.48 ± 0.13 | 0.45 ± 0.10 |
| | Time constant ($\tau$) | –0.06 ± 0.07 | 0.01 ± 0.06 | –0.03 ± 0.06 |
| Predictors | Interaction term ($r{\cdot}\tau$) | 0.20 ± 0.09 | 0.07 ± 0.06 | 0.12 ± 0.09 |

| | | Angular partial correlation coefficients ± standard deviation | | |
|---|---|---|---|---|
| | | **Vestibular** | **Visual** | **Combined** |
| | Angular distance ($\theta$) | 0.57 ± 0.13 | 0.90 ± 0.08 | 0.90 ± 0.06 |
| | Time constant ($\tau$) | –0.06 ± 0.08 | –0.01 ± 0.06 | –0.07 ± 0.06 |
| Predictors | Interaction term ($\theta{\cdot}\tau$) | 0.27 ± 0.11 | 0.28 ± 0.15 | 0.33 ± 0.14 |

