## [Editor Report]

This paper investigates the importance of visual and inertial sensory cues as well as the underlying motion dynamics to the accuracy of spatial navigation. When motion control was artificially manipulated in a virtual environment, subjects could navigate accurately using vision, but not inertial signals alone. Overall, these findings shed new light on how the brain combines sensory information and internal models of control dynamics for self-motion perception and navigation.

---

## [Decision Letter]

**Decision letter after peer review:**

Thank you for submitting your article "Influence of sensory modality and control dynamics on human path integration" for consideration by *eLife*. Your article has been reviewed by 3 peer reviewers, and the evaluation has been overseen by a Reviewing Editor and Richard Ivry as the Senior Editor. The following individuals involved in review of your submission have agreed to reveal their identity: Benjamin Clark (Reviewer #1); Gunnar Blohm (Reviewer #2); Stefan Glasauer (Reviewer #3).

The reviewers have discussed the reviews with one another and the Reviewing Editor has drafted this decision to help you prepare a revised submission.

Summary:

In this manuscript, the authors investigated the importance of visual and vestibular sensory cues and the underlying motion dynamics to the accuracy of spatial navigation by human subjects. A virtual environment coupled with a 6-degrees of motion platform, as described in prior studies, allowed precise control over sensory cues and motion dynamics. To investigate whether control dynamics influence performance, the transfer function between joystick deflection and self-motion velocity was modified at each trial, resulting in subject to rely more on velocity or acceleration to find their way. To explain the main result that navigation error depends on control dynamics, the authors propose a probabilistic model in which an internal estimate of dynamics is biased by a strong prior. Overall, the three reviewers agree this manuscript might be suitable for publication in *eLife* and that additional data are not necessary. However, the analyses need to be clarified and the conclusion better justified. You will find below a summary of the main concerns. Please refer to the reviewers' comments appended at the end for more details.

Essential revisions:

1. Concerns were raised regarding motion cueing that was used to approximate the vestibular cues that would be present during real motion. The reviewers think that it should be better to refrain from generalizing and to restrict the conclusions to this specific artificial type of vestibular input. It could even by interesting, since motion cueing is used in driving simulators. See reviewer #2, point #3 and reviewer #3, point #3.

2. One possible interpretation of the data is that the subjects rely almost exclusively on sensory feedback, and that no estimate of control dynamics is necessary. One caveat of the current design is that the different trial types were interleaved, possibly resulting in unreliable efferent copies (leading subjects to estimate velocity from sensory inputs only) and a history effect in the estimation of tau (biasing vestibular trials). The authors should provide more evidence that their effect is not the result of feedback control only and that there is no history effect. See reviewer #2 point #2 and reviewer #3, point #1-2.

3. The relationship between tau and performance is unclear and should be clarified. Figure 3A seems to contradict Figure 5A. See reviewer #2, point #1.

4. It is unclear why the authors did not propose a more normative framework, e.g. using a hierarchical Bayesian model, as suggested in the discussion. This would be a very interesting addition to the manuscript. See reviewer #2 point #4.

5. The manuscript lack important information and details: number of trials, maximal velocity, difference between males and females, slope of the dependence between time constant and error. The actual control signal, the joystick command, should be shown and analyzed. See reviewer #1, point #1-2; reviewer #3, point #4-5.

6. It seems that tau was correlated with trial duration and velocity (Supp Figure 4), unlike what is stated in the manuscript (the effect of both factors are said to be "unlikely" p 161-167). The author should clarify this point. See reviewer #3, point #5.

7. Data presentation can be improved. See reviewer #1 point #3-5.

*Reviewer #1:*

1) The study tested performance by both male and female subjects. Could the authors comment as to whether sex differences were observed across performance measures? Perhaps sex can be indicated in some of the scatter plots.

2) Figure 2A. It would be helpful if the authors identified the start-point of the trajectory and also provided more explanation of the schematic in the caption.

3) Figure 2B-C. It would be helpful if the authors could expand this section to show some example trajectories and the relationship between examples and plotted data points. This could be done by presenting measures (radial distance, angular eccentricity, grain) for each example trajectory.

4) Because the range of sampled time-constants can vary across subjects, it would nice to show plots as in Figure 3B for each subject (i.e., in supplementary material).

5) Discussion. The broader implications of the findings from the models are not sufficiently discussed. In addition, some comparison could also be made to other recent efforts to model path integration error (e.g., PMC7250899).*Reviewer #2:*

The authors asked how the brain uses different sensory signals to estimate self-motion for path integration in the presence of different movement dynamics. They used a new paradigm to show that path integration based on vision was mostly accurate, but vestibular signals alone led to systematic errors particularly for velocity-based control.

While I really like the general idea and approach, the conclusions of this study hinge on a number of assumptions for which it would be helpful if the authors could provide better justifications. I also have some clarification questions for certain parts of the manuscript.

1) lines 26-7: "performance in all conditions was highly sensitive to the underlying control dynamics". This is hard to really appreciate from the residual error regressions in Figure 3 and seems to be contradicting Figure 5A (for vestibular condition). A more explicit demonstration of how tau affects performance would be helpful.

2) One of the main potential caviats I see in the study design is the fact that trial types (vest, visual, combined) were randomly interleaved. In the combined condition, this could potentially result in a form of calibration of the vestibular signal and/or a better estimate of tau that then is used for a subsequent vestibular-only trial. As such, you'd espect a history effect based on trial type more so (or in addition to) simple sequence effects. This is particularly true since you have a random walk design for across-trial changes of tau. In other words, my question is whether in the vestibular condition participants simple use their previous estimate of tau, since that would be on average close enough to the real tau?

3) I thought the experimental design was very clever, but I was missing some crucial information regarding the design choices and their consequences. First, has there been a psychophysical validation of GIA vs pure inertial acceleration? Second, were GIAs always well above the vestibular motion detection threshold? In other words could the worse performance in the vestibular condition be simply related to signal detection limitations? Third, how often did the motion platform enter the platform motion range limit regime (non-linear portion of sigmoid)?

4) lines 331-345: it's unclear to me why you did not propose a more normative framework as outlined here. Especially, a model that would "contrain the hypothesized brain computationa dn their neurophysiological correlates" would be highly desirable and really strengthen the future impact of this study.

5) I would highly recommend all data to be made available online in the same way as the analysis code has been made available.*Reviewer #3:*

The manuscript describes interesting experimental and modelling results of a novel study of human navigation in virtual space, where participants had to move towards a briefly flashed target using optic flow and/or vestibular cues to infer their trajectory via path integration. To investigate whether control dynamics influence performance, the transfer function between joystick deflection and self-motion velocity was modified trial-by-trial in a clever way. To explain the main result that navigation error depends on control dynamics, the authors propose a probabilistic model in which an internal estimate of dynamics is biased by a strong prior. Even though the paper is clearly written and contains most of the necessary information, the study has several shortcomings, as outlined below, and an important alternative hypothesis has not been considered, so that some of the conclusions are not fully supported by results and modelling.

Substantive concerns

1) The main idea of the paper for explaining the influence of control dynamics is that for accurate path integration performance participants have to estimate dynamics. This idea is apparently inspired by studies on limb motor control. However, tasks in these studies are often ballistic, because durations are short compared to feedback delays. In navigation, this is not the case and participants can therefore rely on feedback control (for another reason, why reliance on sensory feedback in the present study is a good idea, see point 2 below). This means that the task can be solved, even though not perfectly, without actually knowing the control dynamics. Thus, an alternative hypothesis for explaining the results that has not been considered is that the error dependence of control dynamics is a direct consequence of feedback control. Feedback control models have previously been suggested for goal-directed path integration (e.g., Grasso et al., 1999; Glasauer et al., 2007).

To test this assumption, I modelled the experiment assuming a simple bang-bang feedback control that switches at a predefined and constant perceived distance from the target from +1 to -1 and stops when perceived velocity is smaller than an epsilon. Sensory feedback is perceived position, which is assumed to be computed via integration of optic flow. This model predicts a response gain of unity, a strong dependence of error on time constant (slope similar to Figure 3) or of response gain on time constant (Equation 4.1) with regression coefficients of 0.8 and 0.05 (cf. Figure 3D), and a modest correlation between movement duration and time constant (r approximately 0.2, similar to Figure 3A). Thus, a feedback model uninformed about actual motion dynamics and without any attempt to estimate them can explain most features of the data. Modifications (velocity uncertainty, delayed perception, noise on the stopping criterion, etc.) do not change the main features of the simulation results.

Accordingly, since simple feedback control seems to be an alternative to estimating control dynamics in this experiment, the authors’ conclusion in the abstract “that people need an accurate internal model of control dynamics when navigating in volatile environments” is not supported by the current results.

2) Modelling: the main rationale of the model (line 173 ff: “From a normative standpoint, …”) is correct, but an accurate estimate of the dynamics is only required if the uncertainty of the velocity estimate based on the efference copy is not too large. Otherwise, velocity estimation should rely predominantly on sensory input. In my opinion that’s what happens here: due to the trial-by-trial variation in dynamics, estimates based on efference copy are very unreliable (the same command generates a different sensory feedback in each trial), and participants resort to sensory input for velocity estimation. This results in feedback control, which, as mentioned above, seems to be compatible with the results.

3) Motion cueing: Motion cueing can, in the best case, approximate the vestibular cues that would be present during real motion. Furthermore, it is not clear whether the applied tilt is really perceived as linear acceleration, or whether the induced semi-circular canal stimulus is too strong so that subjects experience tilt. Participants might have used the tilt has indicator for onset or offset of translational motion, specifically because it is self-generated, but the contribution of the vestibular cues found in the present experiment might be completely different from what would happen during real movement. Therefore, conclusions about vestibular contributions are not warranted here and cannot solve the questions around “conflicting findings” mentioned in the introduction.

4) Methods: I was not able to find an important piece of information: how many trials were performed in each condition? Without this information, the statistical results are incomplete. It was also not possible to compute the maximal velocity allowed by joystick control, since for Equation 1.9 not just the displacement x and the time constant is required, but also the trial duration T, which is not reported. One can only guess from Figure 1D that vmax is about 50 cm/s for tau=0.6 s and therefore the average T is assumed to be around 8.5 s.

5) Results: information that would useful is not reported. On page 6 it is mentioned that the “effect of control dynamics must be due to either differences in travel duration or velocity profiles”, it is then stated that both is “unlikely”, but no results are given. It turns out that in the supplementary Figure 4A the correlation between time constant and duration/velocity is shown, and apparently the correlation with duration is significant (but small) in the majority of cases. Why is that not discussed in the Results section? Other results are also not reported, for example, what was the slope of the dependence between time constant and error? Why is the actual control signal, the joystick command, not shown and analyzed?

[Editors’ note: further revisions were suggested prior to acceptance, as described below.]

Thank you for resubmitting your work entitled “Influence of sensory modality and control dynamics on human path integration” for further consideration by *eLife*. Your revised article has been evaluated by Richard Ivry (Senior Editor) and a Reviewing Editor.

The manuscript has been improved but reviewer #3 has raised several issues that need to be addressed, as outlined below:

*Reviewer #3:*

The present version of the manuscript has clearly improved, and the authors responded adequately to the comments, and a link to the data was also provided. Some very helpful additional analysis was added, such as shown in Figure 3E. There are, however, some critical points left which are outlined below.

Introduction, line 83f: “These findings suggest that inertial cues alone lack the reliability to support accurate path integration …” Even though in general I’d agree with this statement, the findings in the current paper do not support this claim. Since the inertial cues were generated by motion cueing rather than being natural, it could be that natural inertial cues would yield much better path integration performance. Please change accordingly. See also next comments.

Figure 1 suppl. 2: I agree that the initial tilt cannot contribute to linear path integration, but if it is processed by the central estimator (see, for example, your co-author Jean Laurens’ models), it would change the perceived orientation of the participant to a tilted position. Consequently, the GIA after the tilt would be correctly perceived as being due to tilt, this means it would not be interpreted as resulting from linear displacement, and vestibular input would not at all, or only to a very little part, be used as input to the path integration system. This could be an explanation for the findings of inferior performance in the vestibular condition (see comment above). It would mean that motion cueing as applied here is not appropriate for simulating linear travel, which would be an important finding for designing driving simulators. Please discuss …

Results, page 4: it seems that the fit for the combined condition, specifically for distance (both in terms of R^2^ and of response gain), was worse than for the visual condition. This would be surprising, since adding a second sensory input should not have that effect. However, if the vestibular stimulus, specifically for distance, is not appropriate, then this is exactly what should happen. A conflicting vestibular stimulus could decrease response gain (and the fit).

Results, page 6, line 164ff: “A partial correlation analyses revealed..” A summary statistical result should be shown here as well to support the result of time constant dependence.

Line 165: “…albeit only by modulating the distance dependence” I first misunderstood this and thought it would only modulate radial distance dependence. After looking at Figure 3 suppl 2: maybe better write “…albeit only by modulating both angular and radial distance dependence.”

Figure 5: text in figure caption is missing (probably due to clipping of the text box).

Results page 12-13, Bayesian model: I’m surprised that both SD of likelihood and prior were free parameters. For a Bayes model with Gaussian distributions and fixed prior, only the quotient of both standard deviations is a free parameter (the model is basically equivalent to a weighted sum of the mean of prior and the measurement, with the weight being determined by the quotient of the variances). So, either I misunderstand your model, or there’s a mistake. If the latter is the case, then Figure 6 and the corresponding results are also partly wrong, since likelihood σ and prior σ cannot be determined on their own, but only their quotient. See next comment, I suppose there is really a mistake.

Results page 14, dynamic prior model: here you can easily see from equation 7 (page 25) that there are in fact only 2 free parameters, not three (as you state), if you re-express the weight k: the weight k is given as k=var_p_/(var_p_+var_m_)=1/(1+var_m_/var_p_). So only var_m_/var_p_ is free, not both, you cannot determine both from the fit. Note: in this model, it is usually sufficient to take the first measurement as mean of the first prior (corresponding to a maximum likelihood estimate on the first trial, or uninformative prior). This reduces the model to one free parameter.

Discussion, line 343-344: “In contrast, inertial (vestibular/somatosensory cues) alone lacked the reliability to support accurate path integration …” this is the case for the motion cueing inertial cues, so please make clear here and at other points that your data only refer to this type of inertial cues.

Discussion: I miss a general discussion of the limits of the study due to using motion cueing. As mentioned several times, the results concerning the vestibular and combined conditions of this study cannot be generalized to vestibular stimuli under natural conditions.

Along these lines I’m also very puzzled to read in the authors’ responses the following statement: “Therefore, there is no need to ensure that these accelerations are perceived identically: they *are* identical.”

(This reminds me of an astronaut who once stated that there is no need to study perception of up and down in space, because in weightlessness there is no up and down.)

Two identical linear accelerations can very well be perceived completely differently depending on the rotational history and context. That’s the reason why we perceive a tilt of the head as what it is, and not as rapid linear displacement. Please ask your coauthors Dora Angelaki and Jean Laurens, who are long enough in the field to know this. And this is extremely relevant in the present context.

---

## [Author Response]

Essential revisions:1. Concerns were raised regarding motion cueing that was used to approximate the vestibular cues that would be present during real motion. The reviewers think that it should be better to refrain from generalizing and to restrict the conclusions to this specific artificial type of vestibular input. It could even by interesting, since motion cueing is used in driving simulators. See reviewer #2, point #3 and reviewer #3, point #3.

We agree with the remark and we apologize for our overgeneralization. We have responded to the reviewers’ specific questions about the motion cueing algorithm below. Although we have not claimed to have solved any questions regarding conflicting findings in the literature, we rephrased lines 47-50 such that we do not create the impression that we make such a claim. However, we are confident that our less restricted experimental design generalizes better than the strong vestibular-visual cue conflict that is present when simulating real-world navigation without motion cueing (lines 75-78 and 352-355).

2. One possible interpretation of the data is that the subjects rely almost exclusively on sensory feedback, and that no estimate of control dynamics is necessary. One caveat of the current design is that the different trial types were interleaved, possibly resulting in unreliable efferent copies (leading subjects to estimate velocity from sensory inputs only) and a history effect in the estimation of tau (biasing vestibular trials). The authors should provide more evidence that their effect is not the result of feedback control only and that there is no history effect. See reviewer #2 point #2 and reviewer #3, point #1-2.

History effect: we now test the history effect explicitly. We hypothesize that subjects try to compensate for whatever time constant (tau) they currently believe in, in order to stop at their believed target location. If so, their stopping location should depend only on the target location and not on tau; otherwise they knowingly failed to compensate for tau. In other words, their subjective residual errors (from their believed trajectories) should not depend on tau (correlation equal to zero). If the subjects use their previous tau estimate, then accordingly their believed responses should yield zero correlations between the subjective residual errors and that tau. If participants used the tau from the previous visual/combined trial as their estimate of tau in the current vestibular trial, there should be zero correlation between the tau and the (model-based) subjective residual errors. The data shows that for this model the correlation does not drop to zero when that estimate comes from a visual or combined trial (Figure 7 suppl. 1). In other words, the lack of adaptation cannot be fully explained away by subjects carrying over the tau estimates from the previous combined/visual trials. Intuitively, although the previous tau is, on average, in the direction of the mean of the sampling distribution, it is still far away from the overall mean due to the correlated nature of the random walk. Therefore, carrying over the previous estimate cannot fully explain away how much the participants were oblivious to dynamics changes in the vestibular condition – something that the Bayesian model readily does (Figure 6A). We reference the corresponding figure and analysis in lines 315-323.

Sensory feedback control: With respect to feedback control, we agree that, in principle, it is possible to perform this task suboptimally by directly estimating velocity using sensory feedback in the form of optic flow without estimating the control dynamics. To minimize travel time, bang-bang control is the optimal policy. But to implement it correctly, one must predict accurately when to start braking (switch from +1 to -1) based on the current dynamics. For this reason, we argue that an accurate internal model of the dynamics is required. On the other hand, a bang-bang control policy that is oblivious to the dynamics, such as a purely sensory feedback control model, is guaranteed to result in errors unless the participant performs further corrections. In other words, a bang-bang control policy that is based purely on momentary sensory feedback only allows for inaccurate but never perfect performance.

We appreciate the reviewer’s care in substantiating their idea about feedback control in a simulation of their own! To test whether the reviewer’s model explains our observations, we fit a sensory feedback control model with fixed-distance bang-bang policy to our data with two parameters: mean and standard deviation of switching distance from target (Methods – lines 707-715). The problem with the fixed policy proposed by the reviewer is without adaptation to the control dynamics, the errors have a stronger dependence on those dynamics than we measure experimentally. This fixed model predicts much higher correlations and regressions slopes between the time constant and the residual error than the ones found in the actual data (lines 324-331, Figure 7 suppl. 2A). In fact, the empirical distributions of the switching distances were broad and much closer to that predicted by an ideal bang-bang control policy (that anticipates when to brake using knowledge of the dynamics) than the best-fit fixed-distance bang-bang control policy (Figure 7 suppl. 2C).

We conclude that participants do not fully adapt to the control dynamics, but nor do they ignore them. The previous version of this paper emphasized the imperfection of adaptation, while underemphasizing that there was indeed some significant adaptation. Now we compute the responses expected from both a perfectly adapting policy and a fixed, unadapting one, and show (Figure 3E) that data lies between these extremes, revealing partial adaptation for all conditions. Visual information allows more adaptation, but even in the vestibular condition we still see more adaptation than a fixed controller that does not adapt its policy based on the current dynamics.

3. The relationship between tau and performance is unclear and should be clarified. Figure 3A seems to contradict Figure 5A. See reviewer #2, point #1.

(Please note that Figure 5 has changed to Figure 6 due to our edits). We apologize for the lack of clarity in our description. Under ideal steering control adaptation, stopping positions should depend only on target location, and nothing else (see lines 145-149, 165-168). However, we observed a relationship between the responses and the time constant (Figure 3A), indicating a deviation from ideal adaptation. Our model attributes this deviation to misestimated dynamics (lines 226-228): better/worse control adaptation corresponds to better/worse estimation of the time constant. As pointed out in lines 278-280, better dynamics estimation (Figure 6A; visual/combined) results in smaller modulation of the actual responses by the dynamics (Figure 3A; visual/combined). Conversely, greater misestimation (i.e. vestibular in Figure 6A) leads to stronger modulation (i.e. vestibular in Figure 3A). We rephrased lines 25-27 to better convey the effect of the dynamics on performance. For an explicit demonstration of how the participants’ steering was influenced by the changes in the dynamics, we added lines 185-195 and Figure 4.

4. It is unclear why the authors did not propose a more normative framework, e.g. using a hierarchical Bayesian model, as suggested in the discussion. This would be a very interesting addition to the manuscript. See reviewer #2 point #4.

In this study, we explored the contribution of different sensory modalities, control dynamics, and their interaction in human path integration. The extensiveness of the analyses needed to describe the model performance and beliefs in this novel approach does not leave room for a highly technical and thorough multi-level model of goal-oriented path integration. Such a model should include a detailed description of how participants estimate the dynamics. Our goal was to use this study as a foundation and develop such a model in follow-up studies.

5. The manuscript lack important information and details: number of trials, maximal velocity, difference between males and females, slope of the dependence between time constant and error. The actual control signal, the joystick command, should be shown and analyzed. See reviewer #1, point #1-2; reviewer #3, point #4-5.

We apologize for our omissions. We provided all missing information and additional figures as indicated by the reviewers. We would like to point out that an explicit demonstration of how the participants’ control was influenced by the changes in the dynamics was added in lines 130-134, 185-195 and Figures 2E, 4.

6. It seems that tau was correlated with trial duration and velocity (Supp Figure 4), unlike what is stated in the manuscript (the effect of both factors are said to be “unlikely” p 161-167). The author should clarify this point. See reviewer #3, point #5.

We thank the reviewers for giving us a chance to explain. Prior to the start of data collection, we adjusted stimulus parameters to ensure travel time and mean velocity are similar across different dynamics for a controller with perfect knowledge of the dynamics. However, participants’ knowledge of the dynamics was not perfect, as revealed by the dynamics-dependent responses that we attributed to erroneous adaptation/estimation. Additionally, we found a small dependence of travel duration and average travel velocity on the dynamics for some participants. Importantly, travel duration is a feature of the control policy. Since perfect adaptation would yield identical travel times across dynamics, the resulting dependence on the dynamics again shows that participants failed to adapt perfectly. Simulations confirmed this, as maladaptation results in a dependence of travel duration on the dynamics. We edited lines 207-214 to reflect this explanation and updated Figure 5 suppl. 1 to include our simulations.

7. Data presentation can be improved. See reviewer #1 point #3-5.

Reviewer #1’s comments about data presentation were greatly appreciated. All suggestions were accommodated to the fullest extent we could, and we adjusted other figures in the same spirit (e.g. Figure 1D, 4A).

Reviewer #1:1) The study tested performance by both male and female subjects. Could the authors comment as to whether sex differences were observed across performance measures? Perhaps sex can be indicated in some of the scatter plots.

Since no significant sex differences were observed (see lines mentioned below for stats), we do not indicate the sexes of the participants in the main figures (lines 125-129, 149-152), however, we added Figure 2 suppl. 1B, C to illustrate the difference in performance across sexes.

2) Figure 2A. It would be helpful if the authors identified the start-point of the trajectory and also provided more explanation of the schematic in the caption.

Figure 2A has been updated accordingly and the corresponding caption was expanded.

3) Figure 2B-C. It would be helpful if the authors could expand this section to show some example trajectories and the relationship between examples and plotted data points. This could be done by presenting measures (radial distance, angular eccentricity, grain) for each example trajectory.

We think that the update in Figure 2A addresses the point of the reviewer by identifying the variables in 2B, C as they relate to the trajectory. Because the response gain is not a trial-by-trial measure, it can only be shown as is in Figure 2B, C. Nevertheless, we added Figure 2 suppl. 1A (referenced in line 126) where we display trajectories of an example subject under each sensory condition with the corresponding response gain values displayed on top.

4) Because the range of sampled time-constants can vary across subjects, it would be nice to show plots as in Figure 3B for each subject (i.e., in supplementary material).

The sampling distributions of the time constant across subjects and conditions were added in new Figure 3 suppl. 1, along with plots as in Figure 3B for more subjects.

5) Discussion. The broader implications of the findings from the models are not sufficiently discussed. In addition, some comparison could also be made to other recent efforts to model path integration error (e.g., PMC7250899).

We added a discussion paragraph about the model comparisons and the implications of our findings in lines 378385, while we discuss the proposed study in lines 200-201.

Reviewer #2:The authors asked how the brain uses different sensory signals to estimate self-motion for path integration in the presence of different movement dynamics. They used a new paradigm to show that path integration based on vision was mostly accurate, but vestibular signals alone led to systematic errors particularly for velocity-based control.While I really like the general idea and approach, the conclusions of this study hinge on a number of assumptions for which it would be helpful if the authors could provide better justifications. I also have some clarification questions for certain parts of the manuscript.1) lines 26-7: “performance in all conditions was highly sensitive to the underlying control dynamics”. This is hard to really appreciate from the residual error regressions in Figure 3 and seems to be contradicting Figure 5A (for vestibular condition). A more explicit demonstration of how tau affects performance would be helpful.

We rephrased lines 25-27 to better convey the effect of the control dynamics on performance (i.e. failure to adapt steering to the underlying dynamics). The observed relationship between the responses and the time constant (Figure 3A) denotes a deviation from ideal steering control adaptation (ideal adaptation would manifest as an absence of modulation because stopping positions should depend only on target location and nothing else, see lines 145-149, 165-168). With our model, we attributed this modulation, and thereby, the corresponding erroneous adaptation, to dynamics misestimation (lines 226-228). Therefore, better/worse control adaptation corresponds to better/worse estimation of the time constant. As pointed out in lines 278-280, better dynamics estimation (Figure 6A; visual/combined) results in smaller modulation of the actual responses by the dynamics (Figure 3A; visual/combined). Conversely, greater misestimation (i.e. vestibular in Figure 6A) leads to stronger response modulation (i.e. vestibular in Figure 3A).

Last but not least, for an explicit demonstration of how the participants’ steering was influenced by the changes in dynamics, we added lines 185-195 and Figure 4 describing how participants’ control adapts to changes of the time constant.

2) One of the main potential caviats I see in the study design is the fact that trial types (vest, visual, combined) were randomly interleaved. In the combined condition, this could potentially result in a form of calibration of the vestibular signal and/or a better estimate of tau that then is used for a subsequent vestibular-only trial. As such, you’d expect a history effect based on trial type more so (or in addition to) simple sequence effects. This is particularly true since you have a random walk design for across-trial changes of tau. In other words, my question is whether in the vestibular condition participants simple use their previous estimate of tau, since that would be on average close enough to the real tau?

The motivation for our stimulus design was expressly to provide an opportunity for participants to rely on history. Despite our efforts, we did not find a significant history effect.

As clarified in the response to the question above and in the revised text, the model tries to explain why participants failed to adapt to the changes in tau. In the data, this failure manifests itself as a correlation between response errors and tau (lines 149-155; Figure 3A-D). The proposed model successfully attributes this failure to participants misestimating tau, because response errors obtained by integrating the participants’ control input according to the MAP estimate of tau (instead of the real tau), no longer exhibits such a correlation (Figure 5B; lines 278-280). If alternatively, participants simply used the real tau from the previous visual/combined trial as their estimate of tau in the current vestibular trial, then integrating the participants’ control input using this new way of estimating tau should also result in a correlation of zero.

We tested this and found that, using the previous estimate, model predicted correlations do not drop to zero when that estimate comes from a visual or combined trial (Figure 7 suppl. 1). In other words, the lack of adaptation cannot be fully explained away by subjects carrying over the tau estimates from the previous combined/visual trials. Intuitively, although the previous tau is, on average, in the direction of the mean of the sampling distribution, it is still far away from the mean due to the correlated nature of the random walk. Therefore, carrying over the previous estimate cannot fully explain away the regression towards the mean (the degree to which the participants were oblivious to dynamics changes) – something that the Bayesian model readily does (Figure 6A). We reference the corresponding figure and analysis in lines 315-323.

3) I thought the experimental design was very clever, but I was missing some crucial information regarding the design choices and their consequences. First, has there been a psychophysical validation of GIA vs pure inertial acceleration? Second, were GIAs always well above the vestibular motion detection threshold? In other words could the worse performance in the vestibular condition be simply related to signal detection limitations? Third, how often did the motion platform enter the platform motion range limit regime (non-linear portion of sigmoid)?

1. To determine the parameters of the Motion Cuing Algorithm, we performed several pilot experiments in ourselves to verify the effectiveness of the perception. More detailed answers to the reviewer’s question follow

2. Has there been a psychophysical validation of GIA vs pure inertial acceleration? From the point of view of physics, GIA and pure inertial acceleration are indistinguishable (Einstein, 1907). Therefore, there is no need to ensure that these accelerations are perceived identically: they *are* identical. However, it is possible to sense the rotation movements generated when the MC algorithm tilts the subjects. We analyzed the rotation velocity (Figure 1 suppl. 2B) and found that they could exceed rotation velocity thresholds found in the literature (Lim et al., 2017, MacNeilage et al., 2010) for brief periods (~0.6s), but we argue that these periods are too short to influence our experiment’s outcome (see Figure 1 suppl. 2B).

3. Were GIAs always well above the vestibular motion detection threshold? Yes, we verified (in Figure 1 suppl. 2A) that the GIAs profiles were higher than a conservative motion detection threshold (8 cm/s2, based on Kingma 2005, MacNeilage et al., 2010, Zupan and Merfeld: the thresholds range between 5 and 8.5 cm/s2 in these studies).

4. How often did the motion platform enter the platform motion range limit regime (non-linear portion of sigmoid)? To evaluate this, we show the GIA error in Figure 1 suppl. 2A. Indeed, these GIA error occur if (and only if) the platform is this range limit regime. We show in Figure 1 suppl. 2A that these errors remain well below the GIA threshold: therefore, we don’t think that the limitations of the platform could have influenced the subject’s perception.

4) lines 331-345: it’s unclear to me why you did not propose a more normative framework as outlined here. Especially, a model that would “contrain the hypothesized brain computationa dn their neurophysiological correlates” would be highly desirable and really strengthen the future impact of this study.

In short, this paper is already too dense. We plan to develop such a model, along with behavior and neural recordings in monkeys, which are currently underway. See also response to Reviewing Editor’s comment #4.

5) I would highly recommend all data to be made available online in the same way as the analysis code has been made available.

The dataset was made available online at the following address, and the link was provided in line 719: https://gin.g-node.org/akis_stavropoulos/humans_control_dynamics_sensory_modality_steering

Reviewer #3:The manuscript describes interesting experimental and modelling results of a novel study of human navigation in virtual space, where participants had to move towards a briefly flashed target using optic flow and/or vestibular cues to infer their trajectory via path integration. To investigate whether control dynamics influence performance, the transfer function between joystick deflection and self-motion velocity was modified trial-by-trial in a clever way. To explain the main result that navigation error depends on control dynamics, the authors propose a probabilistic model in which an internal estimate of dynamics is biased by a strong prior. Even though the paper is clearly written and contains most of the necessary information, the study has several shortcomings, as outlined below, and an important alternative hypothesis has not been considered, so that some of the conclusions are not fully supported by results and modelling.Substantive concerns1) The main idea of the paper for explaining the influence of control dynamics is that for accurate path integration performance participants have to estimate dynamics. This idea is apparently inspired by studies on limb motor control. However, tasks in these studies are often ballistic, because durations are short compared to feedback delays. In navigation, this is not the case and participants can therefore rely on feedback control (for another reason, why reliance on sensory feedback in the present study is a good idea, see point 2 below). This means that the task can be solved, even though not perfectly, without actually knowing the control dynamics. Thus, an alternative hypothesis for explaining the results that has not been considered is that the error dependence of control dynamics is a direct consequence of feedback control. Feedback control models have previously been suggested for goal-directed path integration (e.g., Grasso et al., 1999; Glasauer et al., 2007).To test this assumption, I modelled the experiment assuming a simple bang-bang feedback control that switches at a predefined and constant perceived distance from the target from +1 to -1 and stops when perceived velocity is smaller than an epsilon. Sensory feedback is perceived position, which is assumed to be computed via integration of optic flow. This model predicts a response gain of unity, a strong dependence of error on time constant (slope similar to Figure 3) or of response gain on time constant (Equation 4.1) with regression coefficients of 0.8 and 0.05 (cf. Figure 3D), and a modest correlation between movement duration and time constant (r approximately 0.2, similar to Figure 3A). Thus, a feedback model uninformed about actual motion dynamics and without any attempt to estimate them can explain most features of the data. Modifications (velocity uncertainty, delayed perception, noise on the stopping criterion, etc.) do not change the main features of the simulation results.Accordingly, since simple feedback control seems to be an alternative to estimating control dynamics in this experiment, the authors' conclusion in the abstract "that people need an accurate internal model of control dynamics when navigating in volatile environments" is not supported by the current results.

Indeed, in principle, it is possible to perform this task suboptimally by directly estimating velocity using sensory feedback in the form of optic flow, without estimating the control dynamics.

We appreciate the reviewer’s care in substantiating their idea about feedback control in a simulation of their own! To test whether the reviewer’s model explains our observations, we fit a sensory feedback control model with fixed-distance bang-bang policy to our data with two parameters: mean and standard deviation of switching distance from target (Methods – lines 707-715). The problem with the fixed policy proposed by the reviewer is without adaptation to the control dynamics, the errors have a stronger dependence on those dynamics than we measure experimentally. This fixed model predicts much higher correlations and regressions slopes between the time constant and the residual error than the ones found in the actual data (lines 324-331, Figure 7 suppl. 2A). In fact, the empirical distributions of the switching distances were broad and much closer to that predicted by an ideal bang-bang control policy (that anticipates when to brake using knowledge of the dynamics) than the best-fit fixed-distance bang-bang control policy (Figure 7 suppl. 2C).

We conclude that participants do not fully adapt to the control dynamics, but nor do they ignore them. The previous version of this paper emphasized the imperfection of adaptation, while underemphasizing that there was indeed some significant adaptation. Now we compute the responses expected from both a perfectly adapting policy and a fixed, unadapting one, and show (Figure 3E) that data lies between these extremes, revealing partial adaptation for all conditions. Visual information allows more adaptation, but even in the vestibular condition we still see more adaptation than a fixed controller that does not adapt its policy based on the current dynamics.

2) Modelling: the main rationale of the model (line 173 ff: "From a normative standpoint, …") is correct, but an accurate estimate of the dynamics is only required if the uncertainty of the velocity estimate based on the efference copy is not too large. Otherwise, velocity estimation should rely predominantly on sensory input. In my opinion that's what happens here: due to the trial-by-trial variation in dynamics, estimates based on efference copy are very unreliable (the same command generates a different sensory feedback in each trial), and participants resort to sensory input for velocity estimation. This results in feedback control, which, as mentioned above, seems to be compatible with the results.

We manipulated the dynamics in this way precisely because we did not want the efference copy to be fully informative about self-motion velocity. As explained above, we agree that velocity estimation relies predominantly on sensory input. However, as mentioned in the response to the previous question, reliance on sensory input need not necessarily result in pure feedback control, since sensory observations can contribute significantly to the estimation of the control dynamics, which seems to be what the participants are attempting based on our findings (lines 411-412). Pure feedback control would certainly become valuable, and thus much more likely, if we alter the control dynamics within the duration of each trial. This is something that we would like to investigate in future studies.

3) Motion cueing: Motion cueing can, in the best case, approximate the vestibular cues that would be present during real motion. Furthermore, it is not clear whether the applied tilt is really perceived as linear acceleration, or whether the induced semicircular canal stimulus is too strong so that subjects experience tilt. Participants might have used the tilt has indicator for onset or offset of translational motion, specifically because it is self-generated, but the contribution of the vestibular cues found in the present experiment might be completely different from what would happen during real movement. Therefore, conclusions about vestibular contributions are not warranted here and cannot solve the questions around "conflicting findings" mentioned in the introduction.

This comment has also been answered in response to comments of Rev. 2 (see also response to Rev. Editor’s comment #1). Specifically, we have added Figure 1 suppl. 2B (referenced in lines 74-75 and 592-593), where we show the tilt velocity profile over time with a tilt/translation discrimination threshold we chose according to the canal thresholds literature (Lim et al., 2017, MacNeilage et al., 2010). Tilt velocity exceeds the proposed threshold briefly right after trial onset, however, the displacement of the subjects during that period is negligible and should not influence navigation. Thus, perceived tilt can be used as an indicator of trial onset, but it cannot contribute to path integration for 3 reasons: (a) the displacement during that period is negligible, (b) tilt velocity is kept below the perceptual threshold for the remainder of the trajectory, (c) GIA is always above the motion detection threshold of the vestibular system (Figure 1 suppl. 2A).

We have also rephrased lines 47-50 such that we do not create the impression that we make the claim to solve the questions around "conflicting findings". However, we are confident that our relatively less restricted experimental design can be more generalizable when it comes to vestibular contributions in real-world navigation (lines 75-78 and 352-355).

4) Methods: I was not able to find an important piece of information: how many trials were performed in each condition? Without this information, the statistical results are incomplete. It was also not possible to compute the maximal velocity allowed by joystick control, since for Equation 1.9 not just the displacement x and the time constant is required, but also the trial duration T, which is not reported. One can only guess from Figure 1D that vmax is about 50 cm/s for tau=0.6 s and therefore the average T is assumed to be around 8.5 s.

We apologize for our omission. The values for *x* and *T* that we used are added in line 518. Also, we added the number of trials each participant performed in each condition in lines 484-487.

5) Results: information that would useful is not reported. On page 6 it is mentioned that the "effect of control dynamics must be due to either differences in travel duration or velocity profiles", it is then stated that both is "unlikely", but no results are given. It turns out that in the supplementary Figure 4A the correlation between time constant and duration/velocity is shown, and apparently the correlation with duration is significant (but small) in the majority of cases. Why is that not discussed in the Results section? Other results are also not reported, for example, what was the slope of the dependence between time constant and error? Why is the actual control signal, the joystick command, not shown and analyzed?

We thank the reviewer for allowing us to fix these problems. Prior to the start of data collection, we adjusted stimulus parameters to ensure travel time and mean velocity are similar across different dynamics for a controller with perfect knowledge of the dynamics. Nevertheless, participants’ knowledge of the dynamics was incorrect, as revealed by the dynamics-dependent responses that we attributed to erroneous adaptation/estimation. Additionally, we found a small dependence of travel duration and average travel velocity on the dynamics for some participants. Importantly, travel duration is a feature of the control policy. Since, according to our design, perfect adaptation would exhibit similar travel times across dynamics, the resulting dependence on the dynamics merely shows that participants failed to adapt perfectly. Simulations confirmed this, as maladaptation results in a dependence of travel duration on the dynamics.

We edited lines 207-214 to reflect this explanation and updated Figure 5 suppl. 1 to include our simulations. We added the slopes of the regression between the time constant and the residual errors in Figure 3 suppl. 1 and Table 3 (referenced in line 155).

We also added a discussion of analyses (lines 185-195) and figures 1D inset, 2D inset, 4, and Figure 2 suppl. 2 that refer to the joystick input.

[Editors' note: further revisions were suggested prior to acceptance, as described below.]

Reviewer #3:The present version of the manuscript has clearly improved, and the authors responded adequately to the comments, and a link to the data was also provided. Some very helpful additional analysis was added, such as shown in Figure 3E. There are, however, some critical points left which are outlined below.Introduction, line 83f: "These findings suggest that inertial cues alone lack the reliability to support accurate path integration …" Even though in general I'd agree with this statement, the findings in the current paper do not support this claim. Since the inertial cues were generated by motion cueing rather than being natural, it could be that natural inertial cues would yield much better path integration performance. Please change accordingly. See also next comments.

We agree with this comment. We edited this sentence to include this consideration (lines 87-88), and also adjusted the wording in other parts of the text where we refer to inertial cues (lines 358, 381, 412).

Figure 1 suppl. 2: I agree that the initial tilt cannot contribute to linear path integration, but if it is processed by the central estimator (see, for example, your co-author Jean Laurens' models), it would change the perceived orientation of the participant to a tilted position. Consequently, the GIA after the tilt would be correctly perceived as being due to tilt, this means it would not be interpreted as resulting from linear displacement, and vestibular input would not at all, or only to a very little part, be used as input to the path integration system. This could be an explanation for the findings of inferior performance in the vestibular condition (see comment above). It would mean that motion cueing as applied here is not appropriate for simulating linear travel, which would be an important finding for designing driving simulators. Please discuss …

We edited the legend in Figure 1 Suppl. 2 to reflect this view and point to the main text for a more detailed discussion (lines 435-450). We want to point out that, in a previous study, we measured the participants’ performance in complete absence of sensory cues (Lakshminarasimhan, 2018 Figure S1B). Compared to the response measured in the present study using vestibular feedback, we find that the performance is much worse in the absence of sensory feedback, suggesting that participants used the generated vestibular cues to some extent (no sensory cues condition correlation ± SD between target position and response, radial component: 0.39±0.12, angular component: 0.58±0.2; vestibular condition correlation ± SD between target position and response, radial component: 0.64±0.05, angular component: 0.93±0.03; radial component t-test p=9 ∙ 10^−7^, angular component t-test p = 10^−8^). However, we agree that the possibility that the perceived tilt would influence the processing of vestibular inputs cannot be ruled out. We hope that our revised text clearly highlights this limitation.

Results, page 4: it seems that the fit for the combined condition, specifically for distance (both in terms of R^2^ and of response gain), was worse than for the visual condition. This would be surprising, since adding a second sensory input should not have that effect. However, if the vestibular stimulus, specifically for distance, is not appropriate, then this is exactly what should happen. A conflicting vestibular stimulus could decrease response gain (and the fit).

This is a good observation. Although R^2^ and the response gain for the combined condition are slightly lower than the visual condition (only for the radial component), their difference is not significant as R^2^ and response gain for combined falls within the 95% CI of the mean R^2^ and response gain of the visual condition, respectively (Radial component: combined condition mean R^2^ = 0.64, visual condition 95% CI of mean R^2^ = [0.62 0.72], paired t-test p=0.074; Angular component: combined condition mean R^2^ = 0.96, visual condition 95% CI of mean R^2^ = [0.93 0.98], paired t-test p=0.37). Other than the concerns raised in the previous comment about the effect of perceived tilt on path integration and performance, our experimental design should not allow for a sensory conflict. We propose further manipulations for future experiments that would investigate the relationship between vestibular and visual cues in our edited Discussion section (lines 446-450).

Results, page 6, line 164ff: "A partial correlation analyses revealed.." A summary statistical result should be shown here as well to support the result of time constant dependence.

We apologize for this omission. We added Table 4 that shows these values, since the total number of values to be shown (3 conditions x 3 predictors) would take up too much space and would be hard to read. We refer to the Table on line 174.

Line 165: "…albeit only by modulating the distance dependence" I first misunderstood this and thought it would only modulate radial distance dependence. After looking at Figure 3 suppl 2: maybe better write "…albeit only by modulating both angular and radial distance dependence."

We apologize for the confusion, we changed the wording as suggested.

Figure 5: text in figure caption is missing (probably due to clipping of the text box).

Apologies, corrected.

Results page 12-13, Bayesian model: I'm surprised that both SD of likelihood and prior were free parameters. For a Bayes model with Gaussian distributions and fixed prior, only the quotient of both standard deviations is a free parameter (the model is basically equivalent to a weighted sum of the mean of prior and the measurement, with the weight being determined by the quotient of the variances). So, either I misunderstand your model, or there's a mistake. If the latter is the case, then Figure 6 and the corresponding results are also partly wrong, since likelihood σ and prior σ cannot be determined on their own, but only their quotient. See next comment, I suppose there is really a mistake.

Thank you for pointing this out. Indeed, there was a subtle error on our behalf. We corrected and re-fitted our models (static and dynamic prior) with just the ratio *λ* of prior σ over likelihood σ instead of both prior and likelihood σ, and therefore decreased the fitted parameters to 2 and 1 for the static and dynamic prior models, respectively. We updated the text and the relevant statistics and figures.

Results page 14, dynamic prior model: here you can easily see from equation 7 (page 25) that there are in fact only 2 free parameters, not three (as you state), if you re-express the weight k: the weight k is given as k=var_p_/(var_p_+var_m_)=1/(1+var_m_/var_p_). So only var_m_/var_p_ is free, not both, you cannot determine both from the fit. Note: in this model, it is usually sufficient to take the first measurement as mean of the first prior (corresponding to a maximum likelihood estimate on the first trial, or uninformative prior). This reduces the model to one free parameter.

We want to clarify that we use *k* only to update the mean of the prior distribution across trials. After our corrections according to the previous comment, we fit just the ratio *λ* of prior σ over likelihood σ, and set the initial prior to be the time constant on the first trial. This reduced the fitted parameters of the dynamic prior model to 1.

Discussion, line 343-344: "In contrast, inertial (vestibular/somatosensory cues) alone lacked the reliability to support accurate path integration …" this is the case for the motion cueing inertial cues, so please make clear here and at other points that your data only refer to this type of inertial cues.

We already adjusted the wording to this sentence in response to the first comment.

Discussion: I miss a general discussion of the limits of the study due to using motion cueing. As mentioned several times, the results concerning the vestibular and combined conditions of this study cannot be generalized to vestibular stimuli under natural conditions.

We have considered the comments and concerns raised carefully and made the necessary adjustments to the text. As mentioned in the responses above, we clarified wherever applicable that any conclusions made based on our findings apply only to this specific paradigm (lines 87-88, 358, 381, 412). We also added a paragraph in Discussion describing the limitation of the Motion Cueing algorithm and opportunities for future work (435-450).

Along these lines I'm also very puzzled to read in the authors' responses the following statement: "Therefore, there is no need to ensure that these accelerations are perceived identically: they are identical."(This reminds me of an astronaut who once stated that there is no need to study perception of up and down in space, because in weightlessness there is no up and down.)Two identical linear accelerations can very well be perceived completely differently depending on the rotational history and context. That's the reason why we perceive a tilt of the head as what it is, and not as rapid linear displacement. Please ask your coauthors Dora Angelaki and Jean Laurens, who are long enough in the field to know this. And this is extremely relevant in the present context.

All our apologies: this is a misunderstanding. Yes, the combination of rotation and acceleration experienced during tilt can be perceived differently from the acceleration experienced during translation. The misunderstanding originated from the way we think about it: in our mind, it is the rotation history (sensed by the canals) that makes the difference, whereas the accelerations are the same (that is to say, in the absence of rotation sensors, the acceleration induced by tilt and translation are indistinguishable); hence our response.